

# Sensitivity of Gyrescale Marine Connectivity Estimates to Fine-scale Circulation

Saeed Hariri[1,2] , Sabrina Speich[1], Bruno Blanke[3], Marina Lévy[4]

[1] LMD-IPSL, École normale supérieure, PSL,  24 rue Lhomond, 75005 , Paris, Cedex 05, France
[2] Physical Oceanography and Instrumentation, Leibniz Institute for Baltic Sea Research Warnemünde (IOW), Seestraße 15, D-18119, Rostock, Germany
[3] Laboratoire d'Océanographie Physique et Spatiale, UMR 6523, CNRS-IFREMER-IRD-UBO, Brest, France
[4] Sorbonne Université, LOCEAN-IPSL, CNRS/IRD/MNHN, Paris Cedex 05, France.

*Correspondence to*: Saeed Hariri (saeed.hariri@io-warnemuende.de)

**Abstract.** We investigate the connectivity properties between different ocean stations in an idealized open ocean model of a western boundary current system separating two ocean gyres. We applied a Lagrangian framework to compute trajectories from various dynamical setups: a high-resolution (1/9°) 3D velocity field reproducing a large range of the ocean fine-scale (i.e. mesoscale plus part of the submesoscale) dynamics, or a filtered velocity field on a coarse-resolution (1°) grid, and one limited to the surface 2D velocities. As ocean connectivity has been assessed in the published literature using different definitions, in this work we compare different metrics such as the average values of transit time and arrival depth between specified sample stations as well as the probability density functions (PDFs) of transit times and betweenness for the different dynamical setups. Our results indicate that almost none of the PDFs show Gaussian behaviour. When the fine-scale dynamics are taken into account, the numerical particles move and connect pairs of stations faster (between 100 days to 300 days) than when it is absent. This is particularly true, along and near the jets separating the two gyres. Moreover, the
connectivity is facilitated when 3D instead of 2D velocities are considered. Finally, our results suggest that western boundary currents are characterized by high betweenness centrality values, which confirms its key role in controlling the transfer of particles in the double-gyre configuration.

## 1 Introduction

Solutions to a number of problems important to the marine environment require knowledge of connectivity between sites
through advection of water parcels. In general, connectivity is modelled according to a classical advection-diffusion formalism (Roughgarden et al., 1988; Kinlan and Gaines, 2003; Siegel et al., 2003; Largier, 2003) and has been applied for many purposes such as assessing pollutant concentrations between their sources and other regions, managing water quality, planning pollutant release to coastal or offshore waters, assessing the evolution of oil spills or managing activities related to marine ecosystem protection (Mitarai et al., 2009; Fischer et al., 1979; Grant et al., 2005).
Understanding, for example, the persistence of isolated populations and the flow of genetic information is important for scientists and marine ecosystem managers (Treml et al., 2008; Roughgarden et al., 1988; Gaylord and Gaines, 2000; James et



al., 2002; Palumbi, 2003; Trakhtenbrot et al., 2005). Moreover, spatial and temporal patterns in the distribution of marine organisms are strongly influenced by differences or changes in population connectivity (Treml et al., 2008; Levin, 1992; Warner, 1997). Quantifying connectivity is therefore essential for providing solutions for marine management. Within this general framework, connectivity is usually described as the exchange of individuals between marine populations.

In marine connectivity, the rate and the spatial distribution of population spread are determined by larval dispersal (Jonsson and Watson, 2016; Treml et al., 2008; Levins, 1969; Gaines and Lafferty, 1995; Gaylord and Gaines, 2000). Based on this idea, two hypotheses have been put forward to assess population connectivity in ocean ecosystems. A first idea is that "everything is everywhere but the environment selects" (Baas-Becking; 1934; Fenchel and Finlay, 2004; De Wit and Bouvier, 2006) whereas the other concept assumes that regions of the oceans are not so well connected (Martiny et al., 2006; Casteleyn et al., 2010). According to the first hypothesis, the different areas are connected by ocean currents that shape marine communities (De Wit and Bouvier, 2006). In contrast, the second idea leads to different evolutions of marine communities and focuses on qualitative and spatial differences between species (Sáez et al., 2003). However, the work by Jonsson and Watson (2016) suggests that neither of these concepts is entirely accurate. Indeed, microbial communities show strong genetic differentiation at small spatial scales, and that this has been observed throughout the global ocean. Therefore, ocean dynamics, water temperature, salinity and other environmental factors, as well as competency periods, dispersal time, swimming behaviour, and larval characteristics of the species could all have major effects on connectivity between populations (Scheltema, 1986; Veron, 1995; Treml et al., 2008).

The issue of connectivity is best addressed in a Lagrangian framework. Connectivity has traditionally been approached using an advection-diffusion equation, but this assumes uniformity of advection and diffusivity (e.g., Roughgarden et al., 1988; Kinlan and Gaines, 2003; Siegel et al., 2003; Largier, 2003), which is not necessarily verified in the real ocean and calls into question the robustness of the results. However, oceanographers have long tracked surface water parcels using drifting buoys to characterize Lagrangian trajectories and dispersion statistics (e.g., Poulain and Niiler, 1989; Swenson and Niiler, 1996; Dever et al., 1998; LaCasce, 2008; van Sebille et al., 2012; Poulain and Hariri, 2013; Hariri, 2022), and have applied Lagrangian methods in ocean models (e.g., Blanke and Raynaud, 1997; Alberto et al., 2011; Watson et al., 2011; Mora et al., 2012; Hariri et al., 2015; van Sebille et al., 2018; Hariri, 2020). The latter approach allows for the study of all ocean connections in time and space. Furthermore, trajectory dispersion models show the main effects of particle release positions and ocean currents on the strength and persistence of connections between ocean locations (Dong and McWilliams, 2007; Dong et al. 2009; Mitarai et al., 2009). However, population connectivity has mostly been studied using Lagrangian integration of surface ocean currents (e.g., Treml et al., 2008; Mitarai et al., 2009; Jonsson and Watson, 2016).

In recent years, connectivity analysis is a dynamic and rapidly evolving field of research in marine science and oceanography, partly because there is an increasing demand for information on connectivity that informs effective assessment and management of marine resources. The study of connectivity patterns in networks has brought novel insights across diverse fields especially in climate and ocean science. Based on ocean connectivity approach, as a future scenario of increasing temperatures in North-Atlantic waters, and the risk associated with the expansion of harmful benthic





dinoflagellate Ostreopsis cf., Drouet et al. (2021) used Lagrangian particle dispersal simulations to: (i) establish the current colonization of the species in the Bay of Biscay, (ii) assess the spatial connectivity among sampling zones that explain this distribution, and (iii) identify the sentinel zones to monitor future expansion. In their Lagrangian framework, the ocean circulation was generated with the state-of-the-art ocean general circulation model MARS3D (Lazure and Dumas, 2008)

covering the entire Bay of Biscay. They indicated that the innovative association between eDNA monitoring and hydrological connectivity has enabled the identification of sentinel zones to monitor potential future expansion and development of the species in relation to climate changes.

In another research work related to ocean connectivity, Bharti et al. (2022) performed Lagrangian transport simulations and built connectivity networks to understand the patterns of oceanographic connectivity along the coastline. They assessed the

variation in connectivity networks within and between two monsoonal seasons, across El Niño–Southern Oscillation years and for pelagic larval durations up to 50 days. They detected well-connected communities, mapped frequent connectivity breaks and ranked coastal areas by their functional role using network centrality measures.

In order to understand the patterns of habitat connectivity that arise from the movement of multiple species and to highlight the role of regional processes in maintaining local community structure, Cristiani et al. (2022) used a biophysical model to

simulate dispersal, and conducted a network analysis to predict connectivity patterns across scales for the community of invertebrates associated with seagrass habitat in the British Columbia coast of Canada. Their results highlighted the importance of considering marine communities in their broad seascape context, with applications for the prioritization and conservation of habitat that maintains connectivity.

Recently, Hariri et al. (2022) applied hydrodynamic connectivity estimated with a Lagrangian approach to study

relationships between the regional biogeography of Dinophysis species and water mass circulation along the European Atlantic coast. They used different indexes related to connectivity properties such as mean, median, most frequent transit times to illustrate the dispersion pattern of Dinophysis species in the Bay of Biscay (NE Atlantic). Their results showed that by comparison to the connectivity between shelf waters of French Brittany and English Channel waters, a higher connectivity between shelf waters of French Brittany and the Celtic Sea shelf was observed.

Based on Lagrangian integration, different approaches have been used to quantitatively describe the connectivity of different marine areas. These include Lagrangian probability density functions (PDFs) (Mitarai et al., 2009), graph theory (Rossi et al., 2014) and characteristic time scales associated with surface ocean connectivity (Jonsson and Watson, 2016).

Most of these methods are based on the general definition of a "connectivity time," which depends on oceanographic distances and is defined as the mean time required for particles to move from one location to another (Cowen et al., 2007;

Mitarai et al., 2009). However, in the global ocean, mean and median transit times are not well defined because each particle deployed at a given location will reach all other areas of a defined domain if time is sufficient (Jonsson and Watson, 2016).

Jonsson and Watson (2016) proposed to use the "minimum connectivity time" (Min-T), defined as the fastest travel time from source to destination for numerical particles, inferred from a Dijkstra algorithm (1959). This minimum connection time is known to show a better agreement and correspondence with genetic dispersal in marine connectivity (Alberto et al., 2011).



Following this idea, Costa et al. (2017) used graph theory and transfer probabilities to calculate "betweenness", i.e., the shortest paths between different sites within variable ocean dynamics. They applied the Dijkstra (1959) algorithm, which is one of the most commonly used algorithms in graph theory.

The benefit of using the minimum connection time rather than the average transit time has been addressed in some empirical work (e.g., Mora et al., 2012; Mitarai et al., 2009; Döös, 1995; Cowen et al., 2007). Consequently, in our study, we chose to

apply mean and median values of minimum connectivity time for all particles traveling from one given station to another. Furthermore, the dispersion patterns of the numerical trajectories show the main effects of the position of particle releases and ocean circulation on the strength and persistence of connections between station pairs. Specifically, we will provide a complete matrix containing analyses of the mean, median and most frequent values of minimum connection time between each selected area seeded with numerical particles. .

Quantification and analysis of marine connectivity processes require integrated estimation of complex Lagrangian spatial and temporal scales. In this framework, we computed Lagrangian PDFs based on the probability that water parcels move from one location to another during a given time interval to quantify the time scale at which currents at different depth levels connect different regions (e.g., Mitarai et al., 2009).

Estimating connectivity from Lagrangian analysis requires the knowledge of Eulerian velocity fields. In the ocean, such

velocities are either derived from satellite observations of Sea Level Anomalies, from ocean general circulation models, or from ocean reanalyses, which combine the two. The resolution of such products is often insufficient to fully capture the highly dynamical fine-scale portion of the ocean circulation. Also, many studies have limited the implementation of the Lagrangian approach to the surface layer. This can potentially induce strong bias in the estimates of connectivity, to the intense horizontal and vertical circulation associated with ocean mesoscale eddies and jets and with submesoscale features

such as filaments and fronts.

In the present study, we assess how the connectivity properties of typical oceanic flows are affected by the fine-scale circulation, and discuss the challenges facing ocean connectivity estimates due to the spatial resolution (both horizontally and vertically) of the flow field. We focus on mid-latitude open ocean gyres, typical of the subtropical and subpolar oceanic gyres of the North Atlantic, separated by the western boundary current Gulf Stream-North Atlantic drift system, or of the

North Pacific, separated by the Kuroshio-Oyashu, which are regions where fine scales are particularly intense. The impact of the fine-scale circulation is evaluated by comparing connectivity analyses based on full 3D high resolution (1/9°) velocity field with estimates based on coarse resolution (1°) velocity. The impact of the vertical circulation is evaluated by comparing the previous two analysis with a Lagrangian experiment applied to 2D surface velocity. The velocity flow field used in this study was generated with the NEMO ocean model run at 1/54° resolution (Lévy et al., 2010). We carried out offline

Lagrangian transport simulations of numerical particles released in a set of regularly distributed stations in the study region for five years. We use the ARIANE quantitative Lagrangian approach (Blanke and Raynaud, 1997; Blanke et al. 2012) which integrates all spatial scales of the modelled velocity, to better understand how marine connections depend on the wide range of scales of the ocean circulation. Finally, in this paper, by analytically linking ocean connectivity and network theory, we





defined a trajectory-based formulation of betweenness to identify stations that act as focus of congestion, or bottlenecks, in
the basin.

The remainder of this paper is organized as follows: the model data and the method we used to analyze and assess
connectivity are described in Section 2, the results are presented in Section 3, and the discussion and conclusion are
presented in Section 4.

## 2 Data and methods

### 2.1 Data: The ocean circulation model fields

The ocean circulation was generated with the state-of-the-art ocean general circulation model NEMO (Madec et al., 1998).
The model domain is a 2000x3000 km rectangle, 4 km deep, rotated 45°, with closed boundaries. The model was forced at
its surface by prescribed seasonal buoyancy fluxes and winds (Lévy et al., 2012a). The model equations were solved on a
grid with a resolution of 1/54° on the horizontal. This allows the simulation of mesoscale and submesoscale dynamical
structures with an effective resolution close to 1/9° (the smallest size of the structures which are captured by the model
outside the dissipative range, is less than the grid resolution on which model equations are discretized and solved) (Lévy et
al., 2012b). The model grid consists of 30 vertical levels, with thicknesses ranging from 10 m to 20 m in the upper 100 m,
increasing to 300 m at the bottom. The model equations were integrated for 50 years. In this study we used the last five years
of model output, which were saved every two days at the effective model resolution, i.e., on a 1/9° grid.

The time-averaged solution of the model shows two large oceanic gyres, a subtropical gyre in the south with an anticyclonic
circulation, and a subpolar gyre in the north with a cyclonic circulation, separated by a strong zonal jet, and a series of
secondary zonal jets. This horizontal circulation in the surface layers is characteristic of the North Atlantic or North Pacific,
the strong jet being the equivalent of the Gulf Stream or Kuroshio. It should be noted, however, that our domain is smaller
than that of these two ocean basins. The model velocities are highly turbulent, and show strong variability at the daily scale,
and on horizontal scales < 1°. This mesoscale turbulence is characterized by strong jet oscillations, the formation of
secondary jets, eddies and filaments between eddies, and is associated with intense vertical movements.

In order to assess the impact of this fine-scale circulation on connectivity, we filtered this velocity field on a 1° grid to
remove all variations with scales smaller than 1° and compared the connectivity analyses performed with unfiltered
(hereafter high-resolution) and filtered (hereafter coarse resolution) velocities. Filtering was done according to Levy et al.
(2012b), to preserve averaged velocities, and was applied only in space and not in time to conserve seasonal and higher
frequency variations.

Figure 1 shows a snapshot of the surface vorticity and vertical velocity on March 31 of the first year of the simulation. With
full resolution of the velocity field (HR), the flow is organized with a large number of eddies covering a wide range of
scales, displaying filamentary structures resulting from their nonlinear interaction. The more intense small-scale activity





develops in the vicinity of the two jets, the first one located at around 30°N, and the second at 35°N. Fig. 1a illustrates the importance of meso- and submesoscale structures in shaping currents, in setting scales of spatiotemporal variability and dynamical regimes. Importantly, these features are associated with intense vertical currents (Figs. 1c). When these highly turbulent currents are filtered on a coarse resolution grid, the vorticity is smoother and mainly related to the position of the main jets (Figs. 1b,d).

## 2.2 Methods

### 2.2.1 Simulation of trajectories

In this paper, the focus is on the analysis of the ocean connectivity with numerical particles deployed at different defined stations across the double-gyre current configuration. For this purpose, the positions of the numerical particles at each time step (i.e., every day) were calculated using the Lagrangian tool, ARIANE (http://stockage.univ-brest.fr/~grima/Ariane/).

ARIANE is an open-source, off-line three-dimensional Lagrangian particle tracking model written in Fortran, and is compatible with many OGCM outputs. It works by interpolating velocity values to a given particle position using an analytical scheme and advects the particle over a user-defined time step. A description of the algorithm is given by Blanke and Raynaud (1997) and Blanke et al. (2001).

A total of 100,000 particles were deployed in a cylindrical section of 1° radius around each station. We used 1,600,000

particles for each Lagrangian experiment. Based on the Lagrangian tool, ARIANE, particles reaching domain boundaries continue their movement in parallel with the coastal area. The frequency of particle release was specified with random initial times while the minimum duration of trajectory tracking was one year. Particles were released every 1 m from the surface to the base of the mixed layer, yielding a total of 150 release locations over this depth (667 particles per meter) (Fig. 2).

We performed and compared the properties of three sets of Lagrangian trajectories, one performed using the full resolution

of the velocity field in 3D (HR-3D), one performed using the filtered velocity field in 3D (CR-3D), and one using the full resolution surface-only velocity field (HR-SL).

### 2.2.2 Stations specification

Three sample stations were defined along the main jet (-85°~-68° W and 27°~32° N) on the western side of the basin (stations 10, 11 and 12, Figure 3) and two stations were defined upstream of the secondary jet (-81°~-60° W and 33°~35° N)

at locations with lower kinetic energy (stations 5 and 6), in order to study connectivity between different parts of each jet, for example from the tails (ends) of the jets to their heads and back (see Figure 3). In addition, other stations between the jets were selected to calculate connectivity properties (stations 2, 3, 4; 7, 8, 9; 13 and 15). Also, due to the less energetic flow in the diagonal directions of the basin, 5 stations were used to determine the transfer time from north to south and vice versa. Note that stations 1, 3, 5, 8, 12, and 15 are important to study the connectivity properties across the subtropical and subpolar

gyres (Fig. 3).





### 2.2.3 Lagrangian PDF

The process of particle dispersion by turbulent phenomena can be explained and predicted by the Lagrangian PDF approach that was introduced by Taylor (1921). It has been widely used for turbulent flows (e.g., Pope, 1985, 1994, 2000; Mitarai et al., 2003). This method gives the probability that particles have moved from one location to another during a given time interval.

Since the PDF values provide an estimate of the mean dispersion properties of the numerical particles during the integrated times, a correct estimation of the PDF values requires a large number of Lagrangian trajectories (Mitarai et al., 2009), For the purpose of our study, 100,000 particles were assigned to each station. This number was set to have a significant number of particles for the connectivity estimates but was, however, limited to remain computationally manageable . The Lagrangian PDF for each station connected to another station is obtained by:

$$f'_X(\xi;\tau,a) = \frac{1}{N}\sum_{n=1}^{N}\delta\left(X_n(\tau,a)-\xi\right) \tag{1}$$

N is the total number of Lagrangian particles, $\delta$ is the Dirac delta function, $X_n(\tau,a)$ is the position of the nth fluid particle (expressed as a function of the initial position a and advection time $\tau$ ), and $\xi$ is the sample space variable for X.

### 2.2.4 Betweenness

Furthermore, to obtain more information about the connection between the defined stations, the mean, median and minimum values of transit times were calculated. In addition, betweenness centrality, which measures the number of shortest paths between pairs of nodes (stations) that pass through a given node (Freeman, 1977), was used to identify the stations that control the transport within the connectivity network. Betweenness values (based on the work by Costa et al. (2017)) are calculated as follows:

$$BC(k) = \sum_{i \neq k \neq j} \frac{\sigma_{ij}(k)}{\sigma_{ij}} \tag{2}$$

Where $BC(k)$ is the betweenness value of a node k and $\sigma_{ij}(k)$ is the shortest path in the graph that actually passes through k, with $I \neq k \neq j$. It should be mentioned that the betweenness value is normalized by $(N-1)(N-2)$, where N is the number of nodes in the graph, thus $0 \leq BC \leq 1$(Costa et al., 2017).

Costa et al. (2017) also proposed a method for calculating betweenness values that is based on the weight of the edge defined between two nodes i and j:

$$d_{ij} = \log\left(\frac{1}{a_{ij}}\right) \tag{3}$$





Here, instead of the raw transfer probability matrices $a_{ij}$ (Dijkstra., 1959), Costa et al. (2017) suggested implementing $d_{ij}$. In this paper, the two methods are compared.

## 3 Results

### 3.1 Transit times

A quantitative assessment requires some degree of simplification due to the multiple spatial and temporal scales involved. In this framework, it is useful to determine the probability distribution of the numerical particles deployed from the sample stations for different integration times (see Fig. 4 for the results obtained for station 1). After the first week of deployment, the concentration of numerical particles is higher around the starting positions, as expected. After six months, the particles move a short distance from their initial positions and spread over 5~10 degrees of longitude, depending on the flow velocity. When particles are close to strong jets, they disperse very rapidly (2~6 months), whereas in other parts of the basin, due to slower and less energetic velocities, the dispersion occurs over a longer period (1.5 ~ 2 years).

One and a half years after their release, the particles deployed from the subpolar gyre (station 1) have dispersed in the entire subpolar gyre and have also penetrated in the subtropical gyre, along its eastern edge. Regardless of the initial deployment position, 3.5 years after deployment, almost all particles are concentrated along the two intense jets that separate the two gyres (not shown). For particles leaving station 1, the probability that they reach stations 2, 3, and 4 after 2 years, along the basin diagonal, is between 0.2% and 0.8%, and for stations 5, 6, and 14, it is about 0.5%. This means that connectivity between these stations and station 1 is achieved in less than 2 years. A uniform PDF distribution after 2.5 years for the particles from station 1 shows that in less than 900 days they have spread across more than 75% of the basin. We also note that with longer particle lifetimes, the PDFs show similar behaviour compared with the other stations (Fig.4). After 1.5 years, particles are mostly on the eastern side of the basin, moving slowly southward due to less energetic flow in these areas (Figs. 4c, 4d).

In contrast, particles deployed in the main jet (station 10; not shown) remain mostly close or move slowly to the southern basin during all simulation times. This pattern reveals the strong influence of the jets on particle movement. For this case; the PDF has the highest values in the jet area and in the subtropical gyre. After 5 years, the lowest PDF values for particles reaching the jet and the subtropical gyre are associated with particles initially deployed along the western boundary of the subpolar gyre (e.g., station 2). In conclusion, the PDFs show that particles spend long periods of time in the subtropical gyre, indicating that this regional retention by the highly energetic nonlinear ocean dynamics prevent rapid dispersion in all other regions. This significantly increases the mean particle transit times.



## 3.2 Comparison of connectivity properties between HR-3D and CR-3D

### 3.2.1 PDF histogram for HR-3D and CR-3D

Figure 5 shows the PDFs of the transit times of particles traveling between selected stations for HR-3D and CR-3D. The PDFs are not Gaussian and are skewed with a long tail.

Fig. 5a shows the PDFs of the particles deployed at station 1, in the centre of the subpolar gyre, arriving at station 15, in the centre of the subtropical gyre, whereas Fig. 5b shows the reverse connection, i.e., for the particles deployed at station 15 traveling to station 1. For HR-3D the first particles reach station 15 after about 200 days, and most particles reach this station after about 600 days and the latest particles continue to arrive at station 15 after 1600 days. The CR-3D PDF is shifted in time with respect to HR-3D, with particles only reaching station 15 after about 300 days. The width of the CR-3D PDF is

broader than that of the HR-3D, suggesting a larger but slower spread of particles across the domain before reaching station 15. The median transit time from station 1 to station 15 is 751 days, while the minimum transit time in this direction is 201 days. The Lagrangian connections for particles deployed in station 15 and reaching station1 (i.e., a connectivity in the opposite direction as for the previous case) show a longer transit time and a greater spread for both simulations (Fig. 5b). The CR-3D PDF shows an even larger delay in arrival time compared to HR-3D.

For HR-3D, the mean time required for particles to travel along the basin diagonal from the subpolar gyre to the subtropical gyre (i.e., from station 1 to station 15) is about 796 days, and the modal time is 559 days, compared to 989 and 1262 days, respectively, for the reverse connection (i.e., from south to north, station 15 to station 1). This means that the northward movement along the diagonal is faster than the southward movement. The PDFs distributions cover almost the same time range, although the general shape is different. The same transit times (mean and most frequent values) for CR-3D are 891

and 644 from north to south and 1162 days and 1315 from south to north. The minimum time required for particles from south to north (station 15 to station 1) in the HR model is 153 days shorter than in the CR model (201 vs. 355 days).

To compare the Lagrangian connectivity between the most distant stations with the stations closer to each other and within the main jet ([30° N, -85° W], [30° N, -70° W] ,where the mean energy and eddy kinetic energy show the highest values) we computed the transit time statistics between stations 10 and 12. The results are shown in Fig 5c for the direct connection

(station 10 to station 12) and in Fig. 5d for the opposite direction (station 12 to station 10). They suggest that the connection along the eastward jet is faster (as expected): the first and largest number of particles arrive within 10 days in HR-3D, whereas for CR-3D the arrival time of the first particles is longer (40 days) and the PDF distribution is larger.

As foreseen, the intense and highly energetic eastward jet moves the particles very rapidly eastward, although fine-scale circulation (mesoscale eddies and filaments) generated at the edges of the jet disperse the particles that reach station 12

almost continuously (albeit in decreasing numbers) until about 1400 days. The minimum and median transit times for the HR-3D simulation are 11 and 348 days, while these values are larger for CR-3D (64 and 213 days, respectively). The CR-3D velocity field induces slower connections because the peak velocity of the jet is lower and its width larger. The connection





time in the coarser velocity field is relatively continuous until about 1300 days. This can be explained by the particles traveling through the larger-scale recirculation cells of the subpolar and subtropical gyres before reaching station 12.

In contrast, the HR-3D and CR-3D PDFs have a closer shape and distribution for the opposite (westward) connection, with the first particles reaching station 10 from station 12 in less than 50 days and 452 days on average and 546 days in median time (Fig. 5d). The minimum and mean transit times for particles from station 12 to station 10 are longer. The modal value is 260 days, and the median transit time is 398 days. The similarity in connectivity behaviour for the opposite (westward) connection for both velocity fields suggests that the particles move through the mean larger-scale recirculation cells and

follow the common pathways.

The above PDF results for both simulations (HR-3D and CR-3D) clearly show the impact of the ocean fine-scale dynamics which increase the efficiency of the current advection and accelerates the particle motion; in this case, for the CR-3D simulation, the PDFs of transit times are wider with higher mean and minimum transit times due to the lack of resolved turbulent motions.

**3.2.2 Minimum and median transit time as a function of geographical distance**

Fig. 6 shows the minimum and median values of transit time as a function of distance computed in HR-3D for stations along the basin diagonal. The results indicate that with increasing distances, the transit times (minimum and median) increase linearly. For the particles initially deployed from station 1, the results show almost the same behaviour for median and minimum time, except for connections between station 1 and stations 12 and 15. The shortest minimum transit time in a

diagonal direction is from station 8 to station 12 with a value of 2 days. The fastest connection based on median transit times is from station 8 to station 5, with a value of about 95 days. The minimum transit times from south to north and north to south are almost identical (about 200 days). The longest minimum transit time is for the particles moving from station 12 to station 1, 240 days, with a median value of 1109 days. This suggests that the intense fine-scale circulation facilitate connections between station 8 (which is located in the middle of the diagonal transect) and station 12 (at the eastern end of

the main jet) and slows those between stations 8 and 5 (a median transit time of 225 days versus 95 days). In general, the Lagrangian transit times (median and minimum) for a station pair located at the same distance along the basin diagonals differ. Such a difference arises from the complex trajectories followed by the numerical particles and induced by the small-scale simulated dynamics. On the other hand, for stations pairs located at shorter distance (less than 6 degrees), the minimum transit time is less than 55 days, regardless the station location.

To assess more quantitatively the differences in connectivity between HR-3D and CR-3D, Fig. 7 shows the comparison of minimum and median transit times computed for a subset of stations for the two Lagrangian simulations. The results clearly indicate that the minimum and median transit times in HR-3D are significantly lower than for the coarse-resolution configuration. In HR-3D, the resolved nonlinear dynamics induce intense currents, and the particles move much faster than in CR-3D, in particular for the stations located along the two main jets. Fig. 7 suggests that for distant stations, CR-3D will

not provide realistic information about the connection time between stations. The lack of fine-scale motions in the coarse-



resolution simulation leads to significant delays in the advection of numerical particle, especially for areas where mesoscale variability plays an important role in particle displacement. The results obtained for particles deployed from station 15, for short-range connections (distances less than 10 degrees), show a better match for the median transit time for both configurations, HR-3D and CR-3D. Based on a minimum connection time of less than 50 days, there is some convergence

between HR-3D and CR-3D for particles deployed from station 1, whereas large differences arise for distances greater than 6 degrees and for areas that include hotspots of high eddy kinetic energy. In addition, in HR-3D, the particles not only disperse faster but also more uniformly than in CR-3D, which reduces transit times between stations. From south to north along the diagonal, the results of both simulations (median values of transit times) are similar, showing that in this direction particles follow pathways less affected by small-scale ocean instabilities.

### 3.2.3 Examples of depth arrival PDF for HR-3D and CR-3D

Connectivity studies of marine ecosystems commonly integrate Lagrangian trajectories using 2D surface velocity fields because the focus is on passively drifting biological species (plankton, fish larvae, algae …). We test here the robustness of such a strong assumption by integrating Lagrangian particles in a 3D framework: For each station at the initial integration time step, particles are distributed over the water column extending from the surface to the base of the mixed layer (which

can be as deep as 150 m). Then, the particles are advected by the 3D flow, without any depth constraint. In this way, we can test whether particles in the upper ocean remain at the same depth throughout their journey and thus confirm or invalidate the soundness of using 2D and not 3D velocity fields for marine ecosystem connectivity estimates.

Figures 8a and b show the PDFs of the mean arrival depth of particles initially deployed from station 1 and arriving at station 15 (left panel), and in the opposite direction from station 15 to station 1 (right panel). Our analysis indicates that the majority

of the particles in both simulations remain in a depth range smaller than 165 meters without moving much deeper, although the peak is at the deepest distance in the south-north motion for both HR- 3D and CR-3D and it is quite noisy.

The HR-3D and CR-3D PDFs (Figs. 8a, b) indicate that the particles are almost twice as deep in the south-north connection as in the north-south connection. From south to north, a small percentage of the particles reach the bottom layer (more than 450 meters deep) where frictional processes alter the dynamics and thus play an important role in the transit of the numerical

particles. These processes do not appear to play a role in the north-south movement.

The PDF of the mean arrival depth of particles deployed from station 10 to station 12 (Fig.8c) shows that for the trajectories simulated by the high-resolution fields there is a tail that extends down to 175 meters, with a slightly higher percentage of particles in the subsurface layer compared to the CR results, and the particles in the upper layer tend to travel faster than those are at greater depths, as velocities decrease with depth.

On the other hand, although the mean PDF of arrival depth for particles moving from station 10 to station 12 shows the same behavior in the HR and CR 3D velocity fields, in the opposite direction (station 12 to station 10) the PDF distributions for the two simulations are completely different, considering that the distribution is overall flatter in CR-3D than in HR-3D with a long tail extending to 380 meters in depth.





A comparison of the mean arrival depth of the numerical particles deployed from station 12 to station 10 (Fig. 8d) in the HR
case shows that the majority of the particles remain within 50 meters of the surface layer, while in the opposite direction,
some particles move to greater depths, up to 150 meters. Furthermore and as already mentioned, the numerical trajectories
simulated in CR are relatively uniform across the upper layer where they were initially seeded. Indeed, in CR, more than
70% of the particles deployed from station 12 and arriving at station 10 remain close to the surface mixed layer and the
subsurface, where the effects of turbulence at different scales on the numerical particle distribution are more detectable.
Mainly for all cases examined, in HR-3D, the particles tend to remain in the subsurface layer due to the larger effects of
coherent vortices as well as other structures such as filaments and eddies. The PDFs of arrival depth indicate that the
differences between HR-3D and CR-3D are not limited to the arrival time, but are also detectable on different 3D pathways
for each case, resulting in significant changes on the arrival depth. Thus, we can add that the depth results differ depending
on the direction of motion. Also, the peak for all HR-3D cases is at the depth of less than 10 meters except for the south-
north motion, which shows that in this direction, the particles move more in the vertical direction due to weaker stratification
at depth and less turbulence in the surface layer.

### 3.2.4 Mean connection time fields for sample stations

The mean connection times for three stations (north, west, and south of the basin) are shown in Figure 9.

Fig. 9a shows that the particles from station 1 follow pathways that require the longest time to connect to stations in the
subtropical gyre and western boundary current regions, such as stations 10, 11, 12, 13, 15, and 16.

The particles reached a depth of 150 meters from the surface layer near the eastern side of the basin between (-60°~-55° W)
and (37.5°~42.5° N) near station 4, although the transit time from station 1 to station 4 was less than 300 days (not shown).
Particles deployed from station 1, moving from south to north, take almost 500 days, while they take about 400 days to travel
the same distance from north to south. The shortest mean transit times are between stations 1 and 4, and between stations 1
and 3, both of which are less than 350 days, while the longest connections (from station 1) are associated with stations 10
and 11. This was expected since these stations are located along the strong jet. Note that the mean arrival depth for the
shortest transit time is about 35 meters (not shown), while for the longest transit time, the mean arrival depth is over 100
meters below the surface layer. For station 1, the mean transit time is 1.25 times greater in CR-3D than in HR-3D. Figures 9a
and 9d show similar distributions, although the connection times between station 1 and stations between 30° and 40° latitude
and -72.5° and -62.5° longitude differ significantly.

Figure 9b shows the mean arrival time from station 10. As shown in the mean transit time map, a large area connecting the
southwestern region to the north-eastern region has the lowest values. This clearly shows the direct connection of the
particles seeded in the main jet, which travel fast and reach these areas rapidly. For these regions, the mean arrival depth
values were less than 70 meters (not presented here). The longest connection times are associated with stations 1 and 16 for
particles that initially started from station 10. These particles took over 1200 days to arrive north of station 1 and appear to



be in a shadow dynamical region that is not directly connected to the jet. The results are similar in CR-3D for station 10, although the transit time is significantly higher in the coarser simulation than in the finer resolution simulation (Figs. 9b, 9e). Figures 9c and 9f show the mean transit time of particles initially deployed from station 15 to other stations for HR-3D and CR-3D. The distribution is remarkably different. In the CR-3D simulation, the connection is fast in the southernmost region

and does not allow some transit times to be modelled acceptably, such as the motion from station 15 to the areas around station 16, and from station 15 to the northern part of the basin (north of station 1). This figure clearly indicates that the particles in CR-3D move in a less dynamical velocity field, especially for trajectories moving from south to north and from south to west. The shortest connection time from station 15 and station 14 in HR-3D is less than 13 days, while it is 50 days for CR-3D.

Under all simulated conditions, the effects of the highly energetic small scales on particle transit times can be distinguished between stations. As shown in the surface vorticity snapshots in Fig. 1, filamentary structures and small eddies in the jets separating the two gyres and the subpolar region act as transport barriers, for example for particles traveling from station 15 to station 1. Another reason for the increased transit times for particles deployed from station 15 and arriving in the northern part of the basin is that the flow is less energetic.

## 3.3 Connectivity matrices between station pairs

To complete the study, we compared the minimum and median transit times for all defined stations in the basin. To do this, we used the definition of connectivity using graph theory (Treml et al., 2008; Kininmonth et al., 2010a,b; Andrello et al., 2013). This allows for a representation of Lagrangian connections between station pairs in the form of matrices. Indeed, graphs are a mathematical representation of a network of entities (called nodes) linked by pairwise relationships (called

edges). The application of graph theory to the study of marine connectivity typically consists in representing portions of the sea as nodes. Then, the edges between these nodes represent the transfer probabilities between these portions. In our study, the latter are the different stations we choose and their pairwise connections are the edges. The transfer probabilities estimate the physical dispersion of Lagrangian particles, and graph theory is used to define the hydrodynamic provinces. This way of presenting the results allows us to extract more information about the connectivity properties between the different station

pairs.

### 3.3.1 Betweenness centrality

In graph theory, there is a choice for defining transfer probabilities between stations. Here we follow the original approach of Costa et al. (2017) who used the "betweenness centrality", a well-known graph-theory measure, for 32 sites in the Gulf of Lion using 20 different connectivity matrices obtained with Lagrangian simulations. This measure is appropriate for

identifying the most relevant transfer between stations. High values of this measure are commonly assumed to identify stations (nodes) that support the connectivity of the whole network, or to identify migration stepping stones (Treml et al.,





2008). Costa et al.'s (2017) definition of "betweenness" aims to ensure that the shortest path between two stations is the most likely. Therefore, high betweenness is associated with stations through which a high number of probable paths pass.

Recent work by Ser-Giacomi et al. (2021) also introduced a mathematical expression for betweenness relying only on the information provided by simulated trajectories across a generic dynamical system. They called it "Lagrangian betweenness", which is a function of backward- and forward-in-time finite-time Lyapunov exponents (FTLEs). As they mentioned, there is a strong resemblance between the Lagrangian betweenness patterns derived from trajectories and the betweenness calculated using the classical network definition that we used in our paper. Therefore, betweenness uncovers an emerging relationship between the concept of bottlenecks in networks and hyperbolicity in dynamical systems. In addition, Lagrangian betweenness helps to characterize hidden circulation regimes in oceanic flows.

Figure 10 shows the betweenness values for our 16 stations based on 3D Lagrangian simulations calculated with two different methods, node-to-node distances $a_{ij}$ and $\log\left(\frac{1}{a_{ij}}\right)$. A fundamental algorithm in graph theory is Dijkstra's (1959) solution for finding the shortest path between a station and all other stations in a basin, while Costa et al. (2017) made some changes and improvements to it. As mentioned, the main reason for presenting betweenness values is to extract the probability of shortest paths between different nodes on the basis of graph theory. The results show significant differences between the two methods, as also indicated by Costa et al. (2017). The results obtained with $\log\left(\frac{1}{a_{ij}}\right)$ are almost uniform while for the other method, the results show more variable values. This situation mentioned by Costa et al. (2017) is due to changeable conditions in the ocean that lead to implausible connections. The highest betweenness values are associated with stations along both the zonal jets. In addition, the movement from the south and southeast to the southwest part of the basin shows high betweenness values, particularly from stations 13, 14, 15, and 16 to stations 10 and 11. The mentioned condition (high betweenness values) can be adapted for both methods. In general, the results obtained in this study also confirm that the second method shows more probable paths than the first. It can be added that the stations with high mean eddy kinetic energy (along the jets) are special as they are frequently visited by particles connecting with other stations.

The other important outcome of the comparison between PDF plots of transit times and betweenness values is that when the PDF distribution is somewhat symmetrical, the betweenness value is very close to the percentage of particles arriving at the final destination with the mean transit time. When there is no such symmetry, the betweenness values are extremely close to the percentage of particles characterized by the most frequent values of transit time, although in a few cases they are close to the percentage of particles arriving at the median time. In addition, when the distance between the initial station and final destination station is sufficient, the betweenness values are equal to the percentage of particles arriving at the final destination with the mean transit time.



### 3.3.2 Comparison of transit time between station pairs for HR-3D and CR-3D

We evaluate the sensitivity of the connectivity matrix to the currents provided by two different cases: HR-3D and CR-3D. Specifically, in this section, we provide a complete matrix containing the analyses of the median and minimum connection time between each selected area seeded with numerical particles.

Figure 11 provides in an overview of the structure and time characteristics of the connectivity between stations. It shows that the connectivity between the northern and the southern stations is the weakest (the connection is the longest in terms of both the minimum and median time) and it is not symmetric. The longest connection is between the northern and southern stations. The fastest connection is along the main jet (station 10 to station 13). This matrix also highlights the difference in the definition of connectivity when applying the minimum or the median time. The latter is three to four times larger than the first. Moreover, the minimum time for CR is slightly larger than for HR, and varies between 10 days and 4 months. The difference increases notably for the median time, including along the principal jet (with a delay in arrival time ranging from 1 to 6 months).

The longest connection is for particles moving from the northern edge of the subpolar gyre (station 1) to the easternmost region between the two zonal jets separating the gyres (station 9), with minimum and median transit times of 516 and 1131 days respectively for HR. For CR and for the same stations these times increase by an additional 30 and 164.5 days, respectively. On the other hand, the shortest connection is between stations 6 and 5, along the northern zonal jet, where we obtained for HR 1 and 13 days as minimum and median transit times, respectively. Note that this result is related to an increased efficiency of the particle advection due to the resolved small-scale nonlinearities which seems to be particularly active in this part of the basin. The resolved small scales act as stirring structures that accelerate the movement of particles around, for example, the peripheries of mesoscale eddies and along filaments. In addition, areas with longer transit times show larger differences between HR and CR (for example, departures from stations 14,15 and 16 and arrivals at stations 1, 2, and 3).

As a link between connectivity matrices and betweenness values (Figs. 10, 11 and supplementary Fig.1), we should add that high connectivity (betweenness) values are related to stations with lower mean transit times. In other words, studying the betweenness matrix helps to predict and define the lowest connection time between defined stations. Our results show that the connection between station 8 to station 12 in CR-3D has the highest betweenness while the median transit time is 179 days. On the other hand, the connections between the southern stations with station 1 have low betweenness values with high median transit times. Overall, the median transit time matrices in CR-3D have higher values than in HR-3D, while the betweenness matrix values in CR-3D are generally lower than the betweenness values calculated for HR-3D. Moreover, the centre of the matrices in HR-3D, which is associated with connections between stations in the centre of the basin, shows lower median transit times with high betweenness values. It means that these stations play an important role in the connection between other stations distributed in the basin. In general, we can conclude that the betweenness and transit time matrices together provide useful information about the connectivity among different regions of the domain.



### 3.3.3 Comparison of connectivity matrices for HR-3D and HR-SL


To determine whether the Lagrangian properties of oceanic flows can be evaluated in a 2D (limited to the surface layer of the ocean), rather than by including the full 3D framework, we compare connectivity properties between defined stations for 2D and 3D high-resolution simulations (Fig. 12). The results show that the transit times for numerical particles deployed in the surface layer are generally shorter than those for particles that started in deeper layers (3D), although there are some

exceptions such as the motion from station 2 to station 6 and 7: in this direction, a high percentage of particles in the surface layer need a longer transit time to reach the final destination than similar particles in deeper layers. This is due to vertical fluxes associated with the displacement of isopycnals by internal dynamics (e.g., eddy pumping or eddy/eddy interaction). Therefore, areas with similar values of connectivity properties in the 2D and 3D simulations suggest that vertical motions for these regions is not strong enough to add complexity to trajectories.

Although, there are many similarities between the median transit time matrices for the HR-3D and HR-SL cases (Fig. 12), the distribution of the betweenness matrices shows more differences (Fig. 10 and supplementary Fig.1); the main reason is related to vertical movements that increases the connections between stations and thus betweenness values ; in other words, the vertical dimension of the trajectories that exists in HR-3D gives the possibility to establish more pathways between the different areas.. This result provides important insight into connectivity properties in the ocean: while 2D simulations

provide useful information on transit times, it is necessary to understand the rate of connections (betweenness)  using 3D simulations.

## 4 Discussion and conclusion

Lagrangian connectivity using ocean models analyses sets of numerical particle trajectories to identify connecting pathways, as well as time scales and transport between oceanic regions. This is a powerful tool to coherently study the connection

between different areas in the ocean. The current study is one of the first large-scale studies to use high-resolution ocean flow data and particle tracking to describe connectivity patterns in a largescale (although idealized) basin.

In this paper, the focus was on analysing the connectivity of different stations in a double gyre ocean model, using a Lagrangian approach with numerical particles. Sixteen stations were specified and in each station 100,000 particles were used for the numerical analysis. Lagrangian properties such as mean, median and modal transit times were calculated to

examine connectivity properties in the North Atlantic. In addition, the probability-density-functions (PDFs) of transit times and mean arrival depths for different simulations were compared. The analysis used high-resolution 3D velocity fields (HR-3D), or surface velocity (HR-SL), or velocity fields averaged over a coarser resolution grid (CR-3D).

The Lagrangian PDF modelling approach, introduced by Taylor (1921), was implemented for the test stations in all the simulations. Almost none of the PDF distributions showed perfect Gaussian behaviour. The particles have different

trajectories to reach their final destinations due to the small-scale motions induced by the resolution of the   fine-scale



dynamics. The results indicate that particles that remain in the surface layer or near the subsurface layer move faster due to intensified velocities resulting from simulating the finescale circulation. In the deeper parts of the basin, particles need more time to reach their final station, as at depth, the velocity intensity decreases due to the effect of the nonlinear dynamics. This finding is confirmed by comparing the PDF of the 2D surface layer simulation with other simulations.

Fine-scale movements, especially in the upper 50 meters of the surface layer, play an important role in particle motion. The numerical particles in the two simulations (HR-3D and CR-3D) show significantly different PDF distributions, especially for movement from the western part of the basin to the eastern part (e.g., from station 10 to stations 4 and 2). It was also found that the particles transported by the high-resolution velocity fields tend to move to deeper parts of the basin compared to the CR-3D simulation.

In the 5-year simulation based on HR-3D velocity fields, the longest route was obtained for particles deployed from station 9 (in the eastern part of the Western boundary current extension) to station 1 (in the subpolar gyre), with an average transit time of 1145 days. This is due to less energetic flow in the areas close to these stations. In contrast, transit along the principal zonal jet (station 10 to station 11) are among the shortest and fastest routes, with an average transit time of 179 days.

        As expected, the numerical particles remain concentrated around their starting position during the first week after their
deployment. But 3 years after their departure and independently of their initial deployment positions, almost all particles concentrate along the two zonal jets. These jets act as attraction hubs that eventually capture most of the particles. Based on the mean arrival depth at the sample stations, we can see that the particles move toward deeper depths in the interior of the ocean, due to the strong nonlinear velocity fields that develop around these jets on the western side of the basin, with a direct impact on the vertical motion of the numerical particles.

Our results emphasize that because ocean circulation is turbulent at horizontal scales 10-100 km, it is not relevant to assess connectivity properties using climatologies or low-resolution (>100 km) velocity fields; moreover we show that connectivity in the ocean is not 2D but 3D, and that assessments based on 2D fields may alter significatively the results.

        Based on network theory and methods applied by Costa et al. (2017), we defined betweenness centrality, which measures the number of shortest paths between station pairs that pass through a given station. The use of betweenness centrality is a guide
to managing the connectivity analysis. We found that some areas act as stepping stones of connectivity (indicated by high betweenness centrality), such as the stations along the zonal jets. Our results show that high betweenness centrality values, depicting the stations controlling particle transfer within a network, were consistently observed in regions along both jets.

        There are several open and unsolved questions related to the impact of small-scale flow variability on Lagrangian connectivity measures. Lagrangian trajectories simulated with the coarse resolution velocity fields do not sufficiently show
the effect of mesoscale eddies on particle dispersion, which results in unreliable Lagrangian indices (e.g., transit time) compared to estimates based on HR model simulations. The CR ocean flow simulation used in this study, with a spatial resolution of ~10 km, is inadequate to describe the mesoscale circulation. Yet, this fine-scale variability has been shown to significantly shape and change the connectivity of the North Atlantic.




In conclusion, the present study highlights the importance of small-scale variability in determining patterns of connectivity
and provides detailed information on Lagrangian connectivity in the North Atlantic. Our results can guide the spatial scales
at which future OGCMs should be run for reliable connectivity analysis; moreover, for Lagrangian studies, we advocate
refining OGCMs to the appropriate resolution with sufficient spatiotemporal accuracy.

## Authors contribution

SH and SP contributed to the data analysis and preparing the figures. SH, SP, BBL and ML contributed to the manuscript
writing and design of the Lagrangian experiments. All authors reviewed and accepted the final version of the manuscript.

## Funding

This work was supported by the French Government 'Investissements d'Avenir' programmes OCEANOMICS (ANR-11-
BTBR-0008). We also acknowledge the mesoscale calculation server CICLAD (http://ciclad-web. ipsl.jussieu.fr) dedicated
to Institut Pierre Simon Laplace modeling effort for technical and computational support.

## Aknowledgment

The authors are also grateful to Dr. Daniele Iudicone and Dr. Laurent Bopp for their insightful discussions.

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

## Figure captions

**Figure 1:** Snapshots on March 31st of a) Surface vorticity at high resolution (HR), b) Surface vorticity at coarse resolution (CR), c) vertical velocity at 40m  at HR, d) and vertical velocity at 40m at CR.

**Figure 2:** Sample trajectories deployed from station 1 in HR-3D.

**Figure 3:** a) Dispersal of sample trajectories on the surface layer in HR-3D, (b) Module of the annual mean velocity and location of the stations.

**Figure 4:** PDF fields of the position of particles after increasing time intervals, in HR-3D. After 7 days (a), after 180 days (b), after 540 days (c), and after 910 days (d).

**Figure 5:** Comparison of HR-3D (black) and CR-3D (gray) transit time distributions, a) for particles deployed initially from station 1 to station 15, b) from station 15 to station 1, c) from station 10 to station 12, and d) from station 12 to station 10.

**Figure 6:** HR-3D minimum (a) and median (b) transit time against geographical distance. Blue: particles initially deployed from station 1; red:  particles initially deployed from station 3; yellow:  particles initially deployed from station 5; purple:  particles initially deployed from station 8; green:  particles initially deployed from station 12; grey:  particles initially deployed from station 15.

**Figure 7:** Comparison of HR-3D and CR-3D minimum and median transit time, (a, b) Along diagonal direction for particles deployed initially from station 1. (c,d) Along front for particles deployed initially from station 10. (e,f) Along diagonal direction for particles deployed initially from station 15.





**Figure 8:** Comparison of HR-3D (black) et CR-3D (grey) arrival depth distributions, a) for particles deployed initially from station 1 to station 15, b) from station 15 to station 1, c) from station 10 to station 12, and d) from station 12 to station 10.

**Figure 9:** Comparison of HR-3D and CR-3D mean arrival (transit) time, (a, d) for particles deployed initially from station 1, (b, e) from station 10, (c, f) from station 15.

**Figure 10:** Comparison of HR-3D betweenness values calculated based on, a) Dijkstra's solution, and b) improved method by Costa et al. (2017).

**Figure 11:** Comparison of HR-3D and CR-3D minimum and median transit time between station pairs, a) minimum transit time for HR-3D, b) difference between minimum transit time at CR-3D and HR-3D, c) median transit time for HR-3D, and d) median transit time for CR-3D.

**Figure 12:** Comparison of HR-3D and HR-SL minimum and median transit time between station pairs, a) minimum transit time for HR-3D, b) difference between minimum transit time at HR-SL and HR-3D, c) median transit time for HR-3D, and d) median transit time for HR-SL.

## Supplementary Figure

**Figure S1:** Betweenness values calculated for CR-3D and HR-2D based on (a, c) Dijkstra's solution and (b, d) improved method by Costa et al. (2017).

**Figure S2:** Comparison of HR-3D and HR-SL minimum and median transit time, (a, b) Along diagonal direction for particles deployed initially from station 1. (c,d) Along front for particles deployed initially from station 10. (e,f) Along diagonal direction for particles deployed initially from station 15.

45 23



**Figure 1:** Snapshots on March 31$^{st}$ of a) Surface vorticity at high resolution (HR), b) Surface vorticity at coarse resolution (CR), c) vertical velocity at 40m at HR, d) and vertical velocity at 40m at CR.



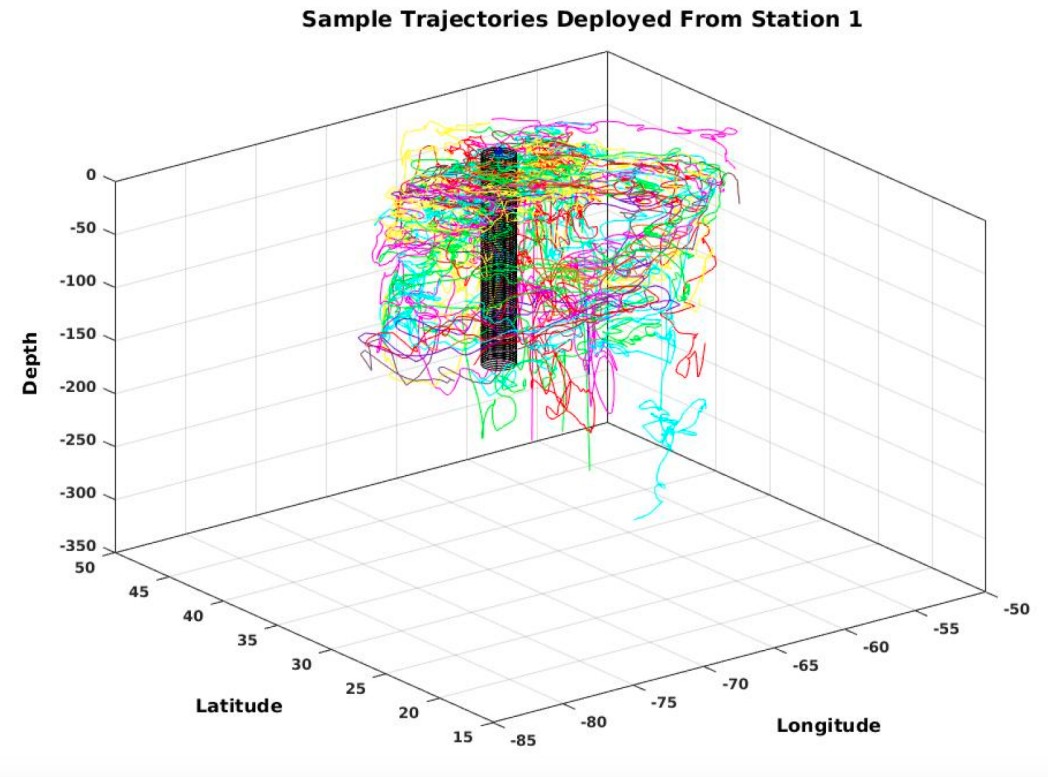

**Figure 2:** Sample trajectories deployed from station 1 in HR-3D.





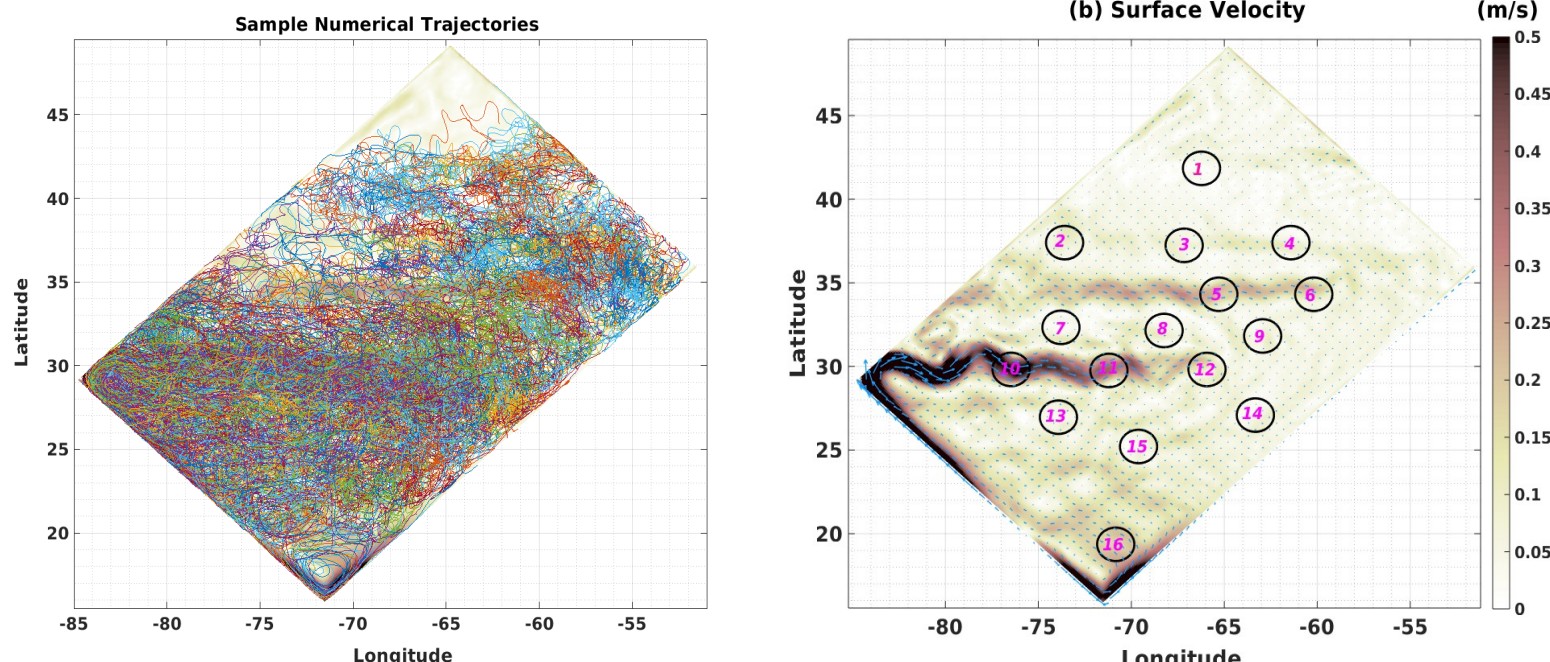

**Figure 3:** a) Dispersal of sample trajectories on the surface layer in HR-3D, (b) Module of the annual mean velocity and location of the stations.





**Figure 4:** PDF fields of the position of particles after increasing time intervals, in HR-3D. After 7 days (a), after 180 days (b), after 540 days (c), and after 910 days (d).




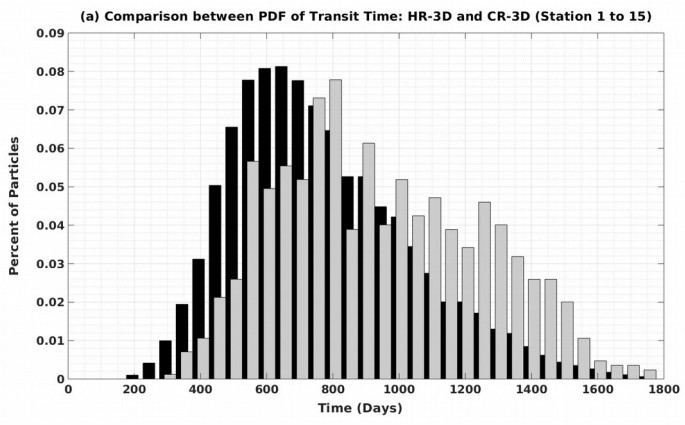
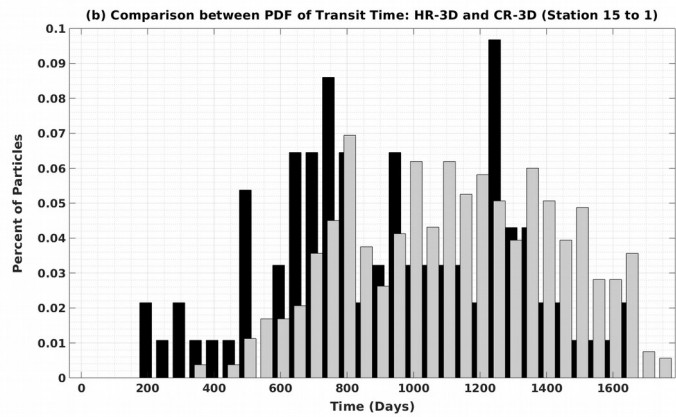

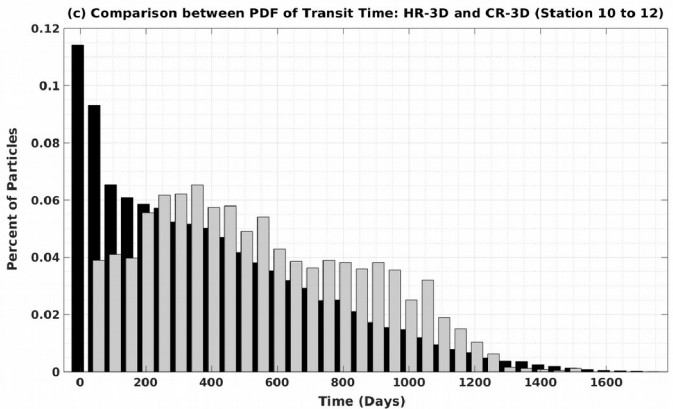
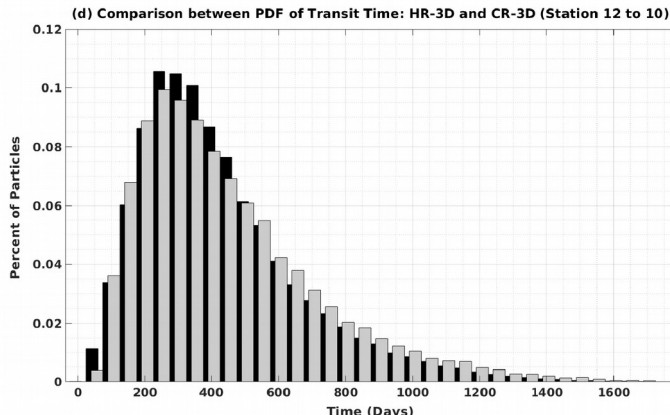

**Figure 5:** Comparison of HR-3D (black) and CR-3D (gray) transit time distributions, a) for particles deployed initially from station 1 to station 15, b) from station 15 to station 1, c) from station 10 to station 12, and d) from station 12 to station 10.





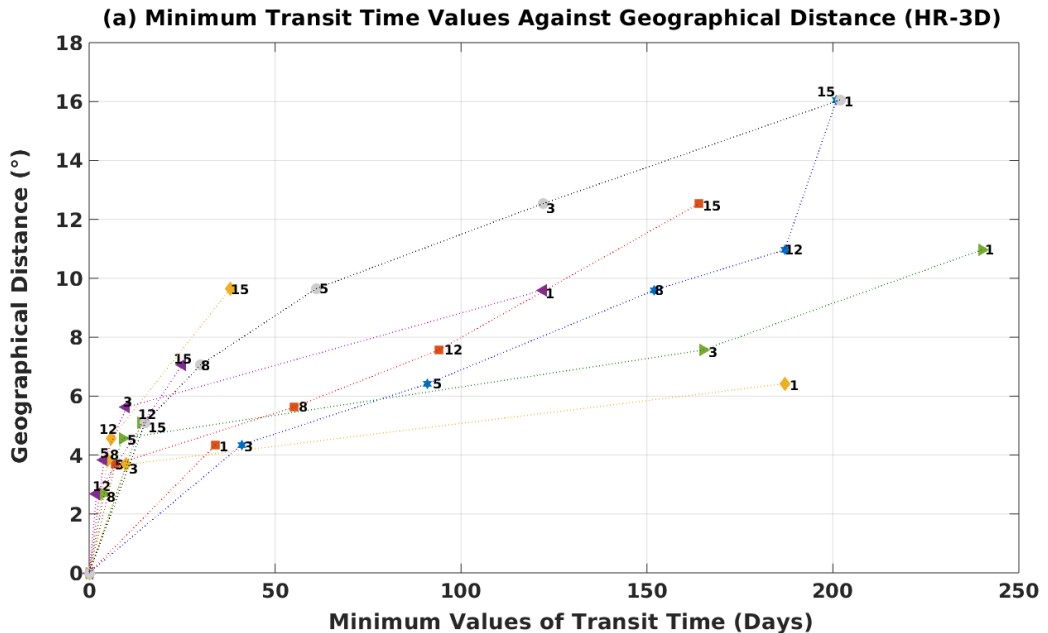

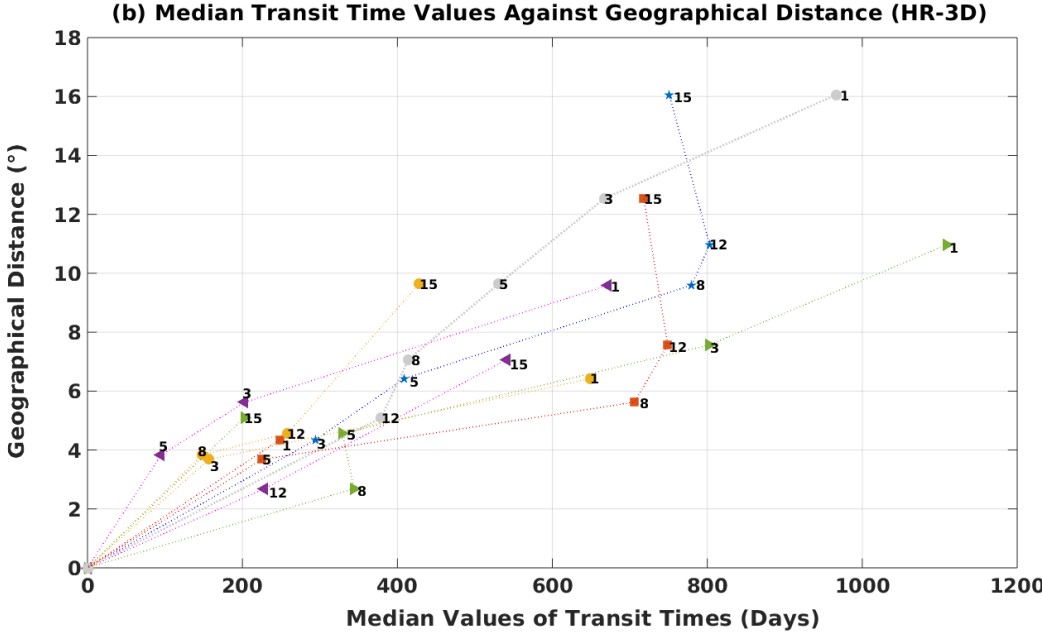

**Figure 6:** HR-3D minimum (a) and median (b) transit time against geographical distance. Blue: particles initially deployed from station 1; red: particles initially deployed from station 3; yellow: particles initially deployed from station 5; purple: particles initially deployed from station 8; green: particles initially deployed from station 12; grey: particles initially deployed from station 15.





**Figure 7:** Comparison of HR-3D and CR-3D minimum and median transit time, (a, b) Along diagonal direction for particles deployed initially from station 1. (c,d) Along front for particles deployed initially from station 10. (e,f) Along diagonal direction for particles deployed initially from station 15.





**Figure 8:** Comparison of HR-3D (black) et CR-3D (grey) arrival depth distributions, a) for particles deployed initially from station 1 to station 15, b) from station 15 to station 1, c) from station 10 to station 12, and d) from station 12 to station 10.





**Figure 9:** Comparison of HR-3D and CR-3D mean arrival (transit) time, (a, d) for particles deployed initially from station 1, (b, e) from station 10, (c, f) from station 15.





**Figure 10:** Comparison of HR-3D betweenness values calculated based on, a) Dijkstra's solution, and b) improved method by Costa et al. (2017).







**Figure 11:** Comparison of HR-3D and CR-3D minimum and median transit time between station pairs, a) minimum transit time for HR-3D, b) difference between minimum transit time at CR-3D and HR-3D, c) median transit time for HR-3D, and d) median transit time for CR-3D.


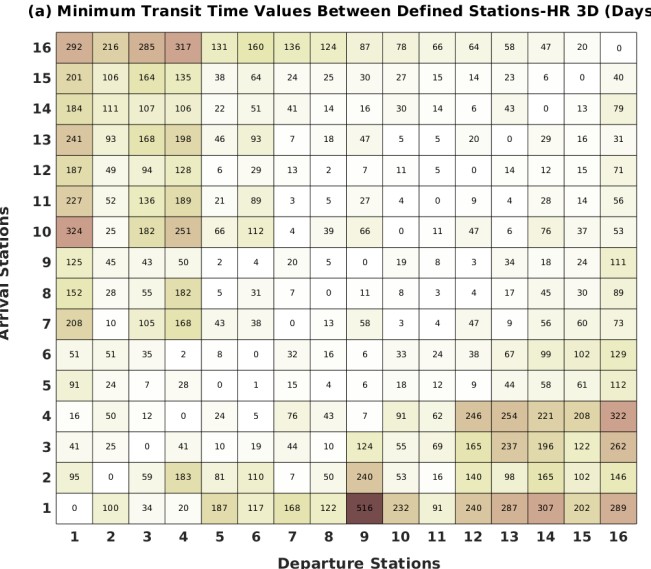

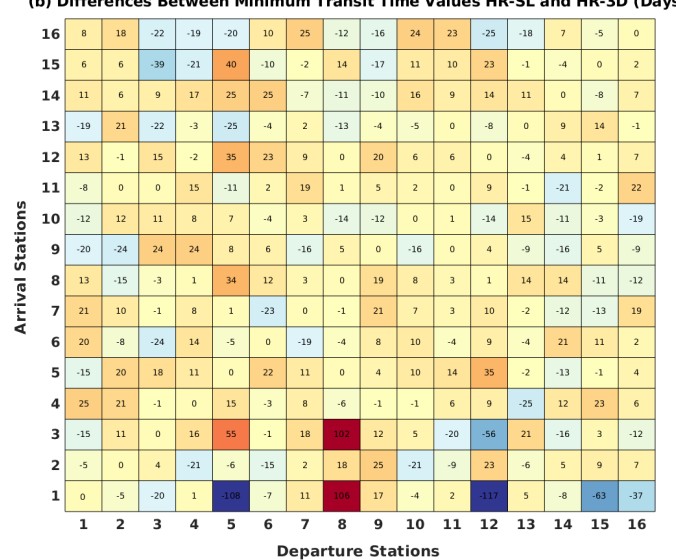

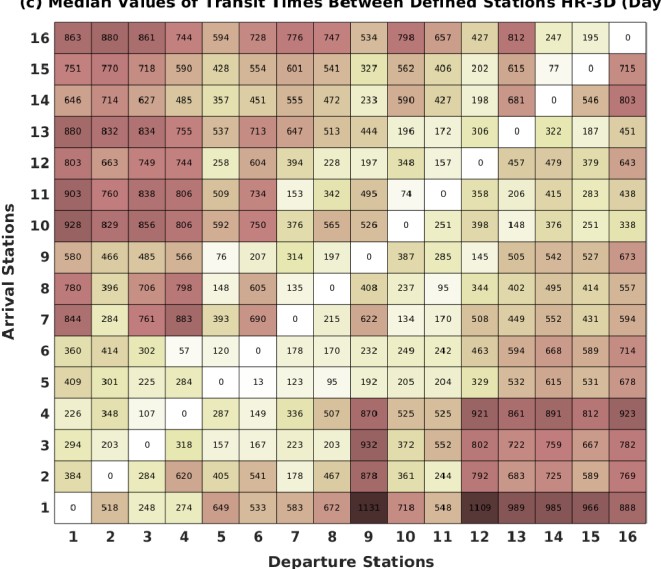

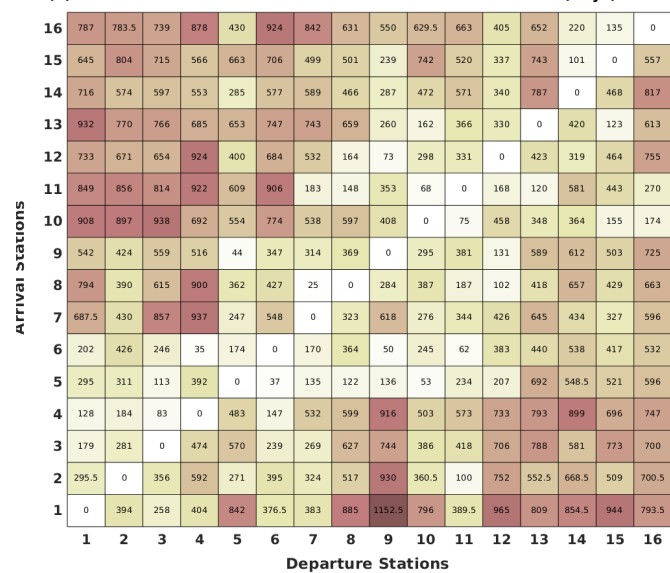

**Figure 12:** Comparison of HR-3D and HR-SL minimum and median transit time between station pairs, a) minimum transit time for HR-3D, b) difference between minimum transit time at HR-SL and HR-3D, c) median transit time for HR-3D,  and d) median transit time for HR-SL.