# Peer review of "Sensitivity of Gyrescale Marine Connectivity Estimates to Fine-scale Circulation"

_EGUsphere, 2022_

## Author Comment (AC1)

We thank Referee #1 for its important and helpful comments. The revised text has been improved using most of them. Here is our specific responses to these comments:
* * *
**C1:** The author define a set of 16 stations across the basin to evaluate connectivity among pairs of them. However, the choice of the number of stations and their location seems arbitrary. How much the analysis is sensitive to such choice? Did the authors tested different stations configurations? Why they did not consider a full covering of the domain instead than a few sparse stations?

**R1:** We thank the reviewer for raising this point. We have expanded the discussion in the paper to justify our approach. There are two aspects to this question. One is the sensitivity of the results to the exact location of the station, in the vicinity of the station. Since there might have been some ambiguity in the meaning of "station" which could have been understood as a single precise location, we now use the word "site" instead of "station" throughout the paper. By sites, we mean small circular regions of 1° radius. This radius corresponds to the largest size of mesoscale eddies. By deploying 100 000 particles at each site, spread at different locations within the site (both on the horizontal and on the vertical), and also at different times, we are able to provide statistical estimates which are reliable at the scale of each site. The second question concerns how different parts of the model domain are connected with one another. To address that, we have considered 16 sites, located in key areas of the model domain, but indeed with some degree of arbitrariness in their exact position. There are 4 key areas in the domain, which are: the subpolar gyre, the subtropical gyre, the jets, and the quieter regions between the jets. To reduce the arbitrariness in the exact position of the sites, we have positioned several sites in each of the key regions. This is now better explained in the text.
* * *
**C2:** A literature review would be useful since the authors did not discuss their work in the context of other similar approaches missing some key references (see specific comments for details).

**R2:** The literature review in the introduction section has been improved by adding more recent studies and references in the field of connectivity analysis. (Please refer to Introduction section).
* * *
**C3:** Regarding the analysis of Lagrangian pdf the authors did not clearly explained when and where they use minimum connection times or connection probabilities and how the two quantities relates between them.

**R3:** The concept of connectivity analysis in this study is based on minimum connection times (please refer to section "2.2.3 Lagrangian indices" in the methodology part). The minimum time required for each particle to reach the destination from the source station was calculated, and further analysis was carried out based on this minimum connection time. Once the minimum

connection time was calculated for each particle, further analysis was carried out. This analysis included the calculation of the mean and median values of the minimum time. These values provide insight into the typical travel time required for particles to reach their destination. Figures [6,7,9,11,12]

In addition, the probability density function (PDF) values were calculated based on the numbers of particles arrived at their destination. This allowed us to understand the distribution of travel times/ arrival depth across the network and identify any patterns or trends in travel times/ arrival depth. These methods have been applied to generate figures [5,8].

**C4:** The authors seem to have misunderstood the concept of betweenness centrality confusing it with the concept of paths across a network (see specific comments).

**R4:** The section on betweenness centrality has been thoroughly revised and improved. The main objective of this paper is to assess the impact of the OGCM resolution and vertical turbulence on the analysis of connectivity in oceanic flows. In addition to this objective, the concept of betweenness was used as a way to investigate the connection between various stations or sites.

In short, we add that betweenness centrality is a way to quantify the importance of a node in a network by measuring how many shortest paths between any two nodes in the network pass through that node. By calculating the betweenness of each node, we can identify the most important locations for water transport in the ocean. Therefore, by comparing the betweenness centrality obtained from different OGCM resolutions, we can simply assess how well the models represent the true connectivity patterns in the ocean, and identify areas for model improvement. (Please refer to Sections 2.2.5 and 3.3 Betweenness Centrality for more information)

*Line-by-line comments on the manuscript:*

**CL1:** The author should also introduce other works where these concepts have been developed, for instance:
- Richter, DJ, et al. "Genomic evidence for global ocean plankton biogeography shaped by large-scale current systems." Elife 11 (2022)
- Ward, BA, et al. "Selective constraints on global plankton dispersal." Proceedings of the National Academy of Sciences 118.10 (2021)
- Jacobi, Martin Nilsson, et al. "Identification of subpopulations from connectivity matrices." Ecography 35.11 (2012)

(l. 93-95) Connection time is just one possible option to characterise connectivity, see for instance different approaches based on fluid fractions (i.e. probabilities):

- Froyland, G, et al. "Almost-invariant sets and invariant manifolds—connecting probabilistic and geometric descriptions of coherent structures in flows." Physica D 238.16 (2009)

- Ser-Giacomi, E, et al. "Explicit and implicit network connectivity: Analytical formulation and application to transport processes." Physical Review E 103.4 (2021)

**RL1:** All of the suggested references have been added to the revised version of paper. ==(Lines. 36, 54, 67, 148).==
* * *
**CL2:** Connection time is just one possible option to characterise connectivity, see for instance different approaches based on fluid fractions (i.e. probabilities)

**RL2:** The aim of this study is to calculate transit time (minimum connection time) and to examine the effects of fine-scale structures on the connectivity properties of the flow. Most of our analysis was based on minimum transit times. However, in addition to this, we used the Lagrangian PDF as a way to analyze the connectivity between different sites during different periods. We added figures based on the PDF fields (==please refer to section 2.2.4 and Fig. 4 and supplementary Figs. S1-5==). Therefore, two different methodologies were used for connectivity analysis: transit (connection) time and Lagrangian PDF.
* * *
**CL3:** (eq. 1) The formula is not explained sufficiently:
- please define the variable "a"
- how a pair of station for which the connectivity is calculated is specified in the equation?

**RL3.** The simplified version of the formula, along with additional information, has been added in the revised version of the paper. (==Line. 173, section 2.2.4 Lagrangian PDF==)
* * *
**CL4:** A general issue along the paper is that the authors did not clearly explained how the connectivity matrix used for network analysis is calculated. Is the matrix defined in terms of times or probabilities? Which algorithm they use to compute its elements?

**RL4:** The matrix used for the network analysis determining betweenness centrality was based on raw transfer probabilities. The calculation of betweenness was carried out using the methods defined by Costa et al. (2017). Instead of using different matrices for each site, the 1,600,000 trajectories deployed from all sites were used together to calculate betweenness values. For each site (node), betweenness values were determined based on the node/edge measure definition. Initially, the raw transfer probabilities $a_{ij}$ were used as edge weights, with the weight decreasing as the probability decreases. However, this method has a drawback, as noted by Costa et al. (2017), in that a high betweenness value could be associated with nodes through which a high number of unlikely paths pass. To address this issue, we applied a new metric ( $d_{ij}$ ), suggested by Costa et al. (2017), which transforms the transfer probabilities $a_{ij}$ into distances .

The connectivity indices presented in Figs. 11-12 are based on the minimum, mean, or median transit time between each pair of stations. To obtain these values, we identified the particles that arrived at the final site (station) from the source site (station) and recorded their arrival time.

In the revised version, we have added a new section "2.2.3 Lagrangian indices" in the methodology part and made the complete revision regarding the betweenness centrality calculation method (section 2.2.5).
* * *
**CL5:** Please note that betweenness centrality and paths-related analysis in fluid flow have been extensively introduced in:
- Ser-Giacomi, E, et al. "Most probable paths in temporal weighted networks: An application to ocean transport." Physical review E 92.1 (2015)
- Lindner, M et al. "Spatio-temporal organization of dynamics in a two-dimensional periodically driven vortex flow: A Lagrangian flow network perspective." Chaos 27.3(2017)

**RL5:** We added these references to the betweenness section. (Line. 183)
* * *
**CL6:** (eq. 2) As commented before, how the matrix elements a_ij are defined? Please note that depending on the definition of the connectivity matrix the distance associated to each step of a path should be evaluated accordingly

**RL6:** As previously replied to **CL4** (**RL4**), the $a_{ij}$ elements are based on the raw transfer probability, and the calculation of the shortest paths involves the sum of a variable number of transfer probability values. However, we have used a different metric, which transforms the transfer probabilities $a_{ij}$ into distances. First, we reversed the order of the probabilities to obtain higher values of the former metric $a_{ij}$. Then, we calculated the log values of the new metric. (Please refer to Sections 2.2.5 and 3.3 Betweenness Centrality for more information)
* * *
**CL7:** This seems an interesting feature? Why such separation is observed?

**RL7:** Particles are subjected to the same forces that drive the movement of water in the gyre. Over time, these particles can be transported by the gyre currents and accumulate in certain areas. This could be due to a number of factors, such as the speed and direction of the currents in these jets, and the interaction between the particles and the water masses in the gyre.
* * *
**CL8:** Why the shape of the pdf is changing qualitatively depending on the velocity fields and/or the pair of stations? Such different features of the pdf should reflect some dynamical proprieties of the advection pattern. Could the authors comment on this?

**RL8.** The shape of the pdf can change qualitatively depending on the velocity fields and the pair of stations because it reflects the dynamical properties of the advection pattern in oceanic flows. The pdf represents the distribution of particles arriving at a destination station from a source station over time, and this distribution is influenced by the complex and variable flow patterns. For example, if the current is fast and unidirectional between two stations, the pdf will likely be narrow and peaked, indicating that particles tend to arrive at the destination station quickly and with little variation in arrival time. In contrast, if the current is slower and more variable, the pdf may be wider and flatter, indicating that particles arrive at the destination station over a broader time range and with more variation in arrival time. Other factors that can influence the shape of the pdf include the presence of eddies or other flow features that cause particles to meander or change direction, as well as variations in source or destination locations that affect the path and travel time of particles.
* * *
**CL9:** This part should probably go to the Methods section and, again, it is not clear how the connectivity matrix is calculated

**RL9:** We have completely revised this part and moved it to the method section related to the calculation of betweenness centrality (Please see sections: 2.2.5 and 3.3 Betweenness centrality).

**CL10:** Please note that the betweenness metric that the authors are trying to calculate is a node and NOT a link propriety! Maybe the authors are confusing the concept of betweenness with the one of a path between a pair of nodes?

**RL10:** We have revised the section on betweenness centrality and highlighted the improved parts on the new version of the paper; we are aware that betweenness is not a link propriety although it is a scalar measure of the number of shortest paths between pairs of nodes that pass through a given node. (Please see sections: 2.2.5 and 3.3 Betweenness centrality)
* * *
**CL11:** No, Costa et al. did not improved the Dijkstra's algorithm.. They used a standard logarithmic transformation for the network links' weights, but they did not change anything of the algorithm.

**RL11:** This was a mistake and we have corrected this section (Please see sections: 2.2.5 and 3.3 Betweenness centrality).
* * *
**CL12:** Again, maybe the authors are confusing the concept of betweenness with the one of path?

**RL12:** We have revised and corrected this section of the paper about the concept of betweenness centrality. (Please see sections: 2.2.5 and 3.3 Betweenness centrality)

---

## Author Comment (AC3)

We thank Referee #3 for its important and helpful comments. The revised text has been improved using most of them. Here is our specific responses to these comments:
* * *
Main issues

**C1.** The definition and interpretation of betweenness and betweenness centrality is wrong in several places in the manuscript. Betweenness is claimed to be used to construct a connectivity matrix, but betweenness is a scalar measure of the number of shortest paths between pairs of nodes that pass through a given node (as correctly mentioned on 212-213). It can be used to identify 'bottlenecks' in the flow (l134): regions through which a relatively large amount of transport occurs (Ser-Giacomi et al., 2021). However, on lines 100-101 it is described as the number of shortest paths between nodes, suggesting that it is a measure defined in matrix form between I and j where it is defined a scalar for each node i. In section 3.3.1, betweenness centrality is wrongly used as a measure of transport probability between nodes, whereas the transport probability should simply be defined from the amount of particles that travels from node i to j (see e.g. Froyland et al., 2014).

Moreover, the paper from Costa et al. (2017) is wrongly interpreted as giving a new definition of betweenness, which, according to the author's is different than that of Dijkstra (1959). Instead, Costa et al. use the textbook definition of betweenness centrality (see Newman 2010) and simply use a reweighting of the edges of the transition matrix that is used as the input graph that betweenness is used on. Dijkstra's algorithm is simply a shortest path computation algorithm, which can still be used, next to Brandes' algorithm for Betweenness computation (Brandes, 2001). So, the "Costa versus Dijkstra" distinction is wrong, but plays a quite central role in this paper.

Moreover, the concept of a 'betweenness matrix' in Figure 10 makes no sense, since betweenness centrality is defined per node, not between nodes. It is therefore unclear what these matrices represent, since it cannot be betweenness centrality. Perhaps transition matrices are really used instead, but then it is unclear which purpose the prior definition of betweenness centrality serves.

These misconceptions should be fully cleared up. This can be done by computing the correct betweenness values per station, which should be in interpreted as "how important is one station as a link between other stations?" and by computing transition matrices, which should be interpreted as "for a given station, what is the probability that it ends up at another station?".

**R1.** We thank the reviewer for raising these points. The section on betweenness centrality has been thoroughly revised and improved. The main objective of this paper is to assess the impact of the OGCM resolution and vertical turbulence on the analysis of connectivity in oceanic flows. In addition to this objective, the concept of betweenness was used as a way to investigate the connection between various stations or sites.

In addition, we have updated the results section by removing all figures related to betweenness centrality and introducing two new figures for a more intuitive and comprehensive explanation of the concept. The new figures provide detailed information on the betweenness values for each node, for a detailed and nuanced understanding of our results (whereas the previous figures were based on the edge betweenness definition).

In short, we add that betweenness centrality is a way to quantify the importance of a node in a network by measuring how many shortest paths between any two nodes in the network pass through that node. By calculating the betweenness of each node, we can identify the most important locations for water transport in the ocean. Therefore, by comparing the betweenness centrality obtained from different OGCM resolutions, we can simply assess how well the models represent the true connectivity patterns in the ocean, and identify areas for model improvement. (Please refer to Sections 2.2.5 and 3.3 Betweenness Centrality for more information)

**C2.** The manuscript uses an idealized two-gyre model representative of a subtropical and subpolar gyre system as found for instance in the North Atlantic. A qualitative interpretation of how the dynamics differ between the HR and CR cases is useful. However, the results section is very lengthy with quantitative descriptions of connectivity properties of different (links between) stations. Since these exact details bear no relevance to the real ocean, the results section can be shortened and sharpened, as only the qualitative results are relevant with respect to our oceans.

**R2.** We removed some unnecessary parts of the text especially by revising the introduction and methodology sections and moreover some parts of the result section.
* * *
**C3.** The authors compare a high-resolution flow field to a coarsened version of it. Then, particle trajectories are integrated on both flow fields. Naturally, particles in the HR case will experience dispersion on scales smaller than the 1-degree grid, which leads to a divergence of the trajectories. I invite the authors to make a remark about the role of this subgrid-scale dispersion and on whether it may be simulated.

**R3.** Please refer to Fig. 1 which present the effects of eddies and also vertical turbulence by comparing the HR and CR models. (Lines. 104~111)
* * *
**C4.** The authors refer to several studies that use Lagrangian PDFs, which usually are PDFs of particle velocities (e.g. Pope, 1985) or particle separation. Please make sure that the referenced papers discuss the type of PDFs used in this manuscript. This will sharpen the definition used. Perhaps using the name Transit Time PDF would already be clearer.

**R4.** It has been corrected. (Lines. 146~147; 166)
* * *
**C5.** The manuscript could benefit from a clearer definition of 'connectivity'. It can help to often plainly talk about transit times or betweenness (if correctly used), as to avoid confusion between the different concepts.

**R5.** The correct definition of connectivity has been added in the revised text (please refer to lines 38-39). Note that in this work we evaluate connectivity in its most general definition of the exchange of particles between different sites instead of one-way transport.
* * *
**C6.** The introduction is unnecessarily lengthy and the discussion of specific papers from line 61-89 is not relevant for the methods and analysis in this paper. For example, the paragraph between 73-77 uses several sentences to mention a study that uses community detection, but community detection is not used in this paper. I see no reason to keep it, as examples of connectivity studies are already mentioned earlier.

**R6.** We revised almost all the main parts of the introduction; in addition, the literature review has been improved by adding more recent studies and references in the field of connectivity analysis (please refer to the Introduction section).

**C7.** The manuscript does not provide any specific hydrodynamic model configuration code, Lagrangian analysis configuration code, or analysis scripts, making it irreproducible. For example, it is unclear how the authors construct the graph/network on which betweenness metrics are computed. Readers would benefit from seeing the code, as the measures such as betweenness are heavily influenced by the way the network is constructed. This omission of code is not in line with the Open Science standards set by EGU journals. Please link to your NEMO model configuration code, Lagrangian simulation scripts, network generation code and analysis code.

**R7.** Sample codes have been added to the paper as supplementary information; these include the method of running the Lagrangian package and other details to be used for the simulation of numerical trajectories.

The NEMO code is available here: https://www.nemo-ocean.eu/
* * *
Line by line comments

- 17: The authors mention that the PDFs are not Gaussian, but this is not a prior hypothesis: there is no reason to assume a Gaussian structure. It is not relevant to mention this, unless it has specific interpretations, which are not given.
  **R:** This sentence has been removed from the abstract.

- 35: Connectivity is not just used in the context of species dispersal, but can also describe the exchange of water (properties) more generally, or that of plastic. See e.g. Froyland et al. (2014) or Ser-Giacomi et al. (2015).
  **R:** It has been added to the introduction section. (Lines 37-38)

- 50: The authors mention that using an advection-diffusion equation assumes uniformity in advection and diffusion coefficients. This is not an inherent assumption in using an advection-diffusion equation, but simply an assumption that is used in the studies mentioned. After all, one can study connectivity using tracers in an OGCM, where advection nor diffusion need to be uniform.
  **R:** This part has been removed from the revised text.

- 57: What is meant by an "ocean connection"?
  **R:** In Lagrangian analysis, the "ocean connection" refers to the study of how particles, such as plankton or pollutants, are transported and dispersed in the ocean. Note that this part has been removed from the introduction section.

- 59-60: "Population connectivity has mostly been studied using Lagrangian integration of surface ocean currents". This is what the authors also do. Currently, the sentence hints at the manuscript providing an alternative method, which it does not.
  **R:** In addition to the 2D connectivity analysis, we also conducted a thorough 3D connectivity analysis for the HR and CR cases. (Please refer to the aim of this study: lines 56~64), although this part has been removed from the introduction section and we are not addressing the case of marine populations directly here. There is no link between this hypothesis and the rest of the paper.

- 93: The concept of a connectivity time is ill-defined and not used by all of the aforementioned examples, such as Rossi et al. (2014). Please provide a precise definition.

  **R:** In the revised draft, this paragraph has been removed from the introduction, and part of it has been moved to the methodology section with corrections. (Section 2.2.3 Lagrangian indices); in general connectivity time in the ocean refers to the time it takes for water masses or particles to travel between two sites/ points in the ocean. This can be important for understanding the transport of nutrients, pollutants, larval marine organisms, and other materials in the ocean.

- 101: This is an example of an incorrect interpretation of betweenness.

  **R:** This part has been removed.

- 103: This implies that Dijkstra's algorithm is for betweenness computation, but Dijkstra (1959) simply concerns a shortest path algorithm.

  **R:** We corrected it. (Please see sections: 2.2.5 and 3.3 Betweenness centrality)

- 107-109: Mention what the aim is of constructing such a matrix. How will it benefit your analysis?

  **R:** We have improved this part and moved it to the methodology section with corrections. (Lines. 158-159)

- 132: "all spatial scales of the modelled velocity". Do other Lagrangian codes not integrate over all spatial scales and just some instead?

  **R:** It depends on the type of depth level used in the OGCM; some packages cannot do 3D integration on time-varying depth coordinates (as it is the case with ROMS/POM/GETM; where "Z" changes at each time step and location "z = z(i,j,k) 3D matrix"). Note that for the NEMO model we did not have this problem.

- 132-133: from "to better understand" is unnecessary.

  **R:** It has been removed from the sentence. (Line. 65)

- 171: Please mention over what integration time particles are integrated.

  **R:** It has been added to the text. (Line. 128)

- 181: It is unclear how often particles are released. Is each particle released at a different initial time-step, or is this done in batches (of which size)?

  **R:** It was mentioned in line 127 ; random initial time step (between the first day of the first year and the last day of the fourth year).

- 183: Are particles also released up to 150m deep if the mixed layer is shallower than that?

  **R:** Yes, particles are also released up to 150m deep.

- 187: The specification of the stations still seems arbitrary to me. How are they chosen exactly?

  **R:** We have expanded the discussion in the paper to justify our approach. There are two aspects to this question. The first is the sensitivity of the results to the exact location of the station, in the vicinity of the station. Since there might have been some ambiguity in the meaning of "station" which could have been understood as a single precise location, we now use the word "site" throughout the paper. By sites, we mean small circular regions of 1° radius. This radius is an upper bound on the largest size of mesoscale eddies. By deploying 100 000 particles at each site, spread at different locations within the site (both on the horizontal and on the vertical), and also at different times, we provide reliable statistical estimates at the scale of each site. The second aspect concerns how the different parts of the model domain are connected with one another. To address this, we considered 16 sites, located in key areas of the model domain, but indeed with a

degree of arbitrariness in their exact position. There are 4 key areas in the domain, which are the subpolar gyre, the subtropical gyre, the jets, and the less turbulent regions between the jets. To reduce the arbitrariness of the exact position of the sites, we have positioned several sites in each of the key regions. This is now better explained in the text.

- 193: "5 stations were used". Which?

  **R:** 1, 3, 8, 15, and 16, information added to the text. (Lines. 141-142)

- 194: "Note that stations 1, 3, 5, 8, 12, and 15 are important". Please explain why.

  **R:** We have deleted this sentence as it was no longer necessary.

- 198: Taylor (1921) is seminal to the theory of eddy diffusion, but unrelated to Lagrangian PDFs.

  **R:** The reference has been removed

- 208: Please define the meaning of the 'sample space variable'.

  **R:** The simplified version of the formula, along with additional information, was added in the revised version of the paper. (Please see **eq. 1**, section 2.2.4 Lagrangian PDF)

- 214: The authors give a textbook definition of betweenness, which was not defined by Costa et al. (2017).

  **R:** It has been corrected. (please see eq. 2 Lines 204, 205 and 206)

- 216: sigma is the sum of shortest paths, not just the shortest path.

  **R:** It has been corrected. (Line. 206)

- 220-221: Costa et al. (2017) only proposed redefining the weights of the Lagrangian transition matrix. Furthermore, please explain what aij is, what dij is, and how these are used to compute betweenness in eq (2).

  **R:** This section has been completely revised. Please see section "2.2.5 Betweenness centrality".

- 224: This comparison is only in the supporting information. I suggest to either remove this sentence or to move the comparison to the main text.

  **R:** It has been removed. Please see section "2.2.5 Betweenness".

- 226-250: Since the paper is mainly about a comparison between HR and CR, the section where transit times are just reported for HR seems irrelevant to me, especially since this quantitative assessment cannot be translated to the real ocean. Instead, qualitative assessments provide a more powerful analysis as these likely commute with the real ocean.

  **R:** This part of the paper presents the application of Lagrangian PDF in connectivity analysis, as also mentioned by referee #1. Although it presents the results related to the HR model (as our reference model), it can help the readers to follow the dispersion of particles during different periods and as a way to learn more about the connection between different sites.

- 232-233: These conclusions cannot be drawn simply from Figure 4 alone, as this concerns only one release site.

  **R:** We agree with the reviewer as Fig. 4 shows the dispersion of the particles during different time periods. For this reason, we added other results as supplementary figures (Figure S1~S5).

- 237: "not shown" --> please show this

  **R:** It is shown as a supplementary figures in the revised version of the paper. (Line. 239)

- 244: "not shown" --> please show this.

  **R:** It is shown as a supplementary figure in the revised version of the paper. (Line. 245)

- 244-245: Do you have a hypothesis for why this behavior is so different from particles release in station one? After all, those particles cross the main jet too. Why are they not trapped in it?

  **R:** Particles are subjected to the same forces that drive the movement of water in the gyre. Over time, these particles can be transported by the gyre currents and accumulate in certain areas. This could be due to a number of factors, such as the speed and direction of the currents in these jets, and the interaction between the particles and the water masses in the gyre.

- 248-250: This seems to be at odds with what happens for the particles in station 1, which can cross the main jet.

  **R:** The results show that the particles deployed from station 2 are nearly six months behind those released from station 1 due to the different types of currents around the stations. This indicates a significant difference in the particle dispersion rate.

- 252-294: This section can be shortened. Details about the distributions are not relevant; only the comparison between the distributions among the CR and HR case are.

  **R:** We removed some unnecessary parts of the text especially by revising the introduction and methodology sections and moreover some parts of the result section.

- 253-254: I don't think Gaussian shapes should be expected in any case. See Van Sebille (2011) or O'Malley (2021) for similar transit time PDFs.

  **R:** We removed this sentence.

- 285: specify "closer"

  **R:** We meant "similar" (Line. 285)

- 330-332: In the case that particles are assumed buoyant, the 2D assumption is still valid. Only for non-buoyant particles, a simplification using only surface currents may be problematic.

  **R:** We agree with the reviewer: the buoyancy of particles can influence the movement of water masses in the ocean, as particles that are more buoyant will be more affected by surface currents and wind-driven mixing, while particles that are less buoyant will be more influenced by deep ocean currents. This can lead to changes in the flow patterns that determine the connectivity between different regions, and can therefore affect the accuracy of connectivity models.

- 363: It is unclear how this figure is plotted. Is any smoothing used? Are all values statistically significant? Is there enough data in each region?

  **R:** To generate this type of figure, we divided the basin into very small bins (1 $km^2$) and calculated the mean arrival time based on the number of particles in the bin. Near the boundaries, we applied some smoothing using flat shading in MATLAB.

- 371: Please also list the longest transit times associated to those stations.

  **R:** It has been added (LineS 368-369)

- 371-373: The authors mention that for the mean arrival depth for the shortest and the longest arrival time differ by about 65 meters. Is this result generalizable? I.e. are short arrival times usually associated with shallower depths? If this is not a generalizable result, it can be left out, since it would be anecdotal.

  **R:** It is more reasonable to have a shorter transit time at shallower depths and this result (difference of about 65 meters) is for one of the sample stations.

- 373-375: Is this a general result or is it anecdotal for this case?

  **R:** As a sample case it is valid for this station.

- 403: The authors mention that graph theory is used to define hydrodynamic provinces, but this concept (Rossi et al., 2014) is not actually used in this paper.

  **R:** We removed the reference and this part has been moved to the methodology section as also suggested by Referee #1. Please see section "2.2.5 Betweenness centrality".

- 407-408: Betweenness centrality is not at all a measure of transfer probabilities between two stations. That measure should simply be the amount of particles traveling between stations over time.

  **R:** We corrected the results and descriptions related to the betweenness section. Please see sections 2.2.5 and 3.3.

- 412: Costa et al. (2014) do not have a special definition of betweenness.

  **R:** We made the correction. Please see sections 2.2.5 and 3.3.

- 414-416: This is not a different type of betweenness, but a different type of graph used to compute it.

  **R:** We made the correction. Please see sections 2.2.5 and 3.3.

- 422: $a_{ij}$ and $\log(1/a_{ij})$ are not distances but weights.

  **R:** We made the correction. Please see sections 2.2.5 and 3.3.

- 435-437: Please show this claim.

  **R:** We removed this part based on the new results obtained for betweenness.

- 443: Renaming "connectivity matrix" to "transit time matrix" avoids confusion.

  **R:** It has been done. (line. 414)

- 465: "High connectivity" and "betweenness" are not the same

  **R:** This part has been removed from the revised version due to the new results provided for betweenness.

- 485-492: This section could use a stronger conclusion drawn. Currently the conclusion is simply that are differences in transit time between the two runs, but it remains unexplained what the precise reason for this is.

  **R:** We added more information. (Line. 452-454)

- 490: Betweenness is not a 'rate of connections'.

    **R:** It has been corrected. Please see section 3.3.

- 495-496: The authors wrongly claim novelty here about using HR flow fields to describe connectivity patterns in a large-scale basin. Rossi et al. (2014) do the same for the Mediterranean and Reijnders et al. (2021) for the Arctic, which are both not idealized.

    **R:** We mentioned it because it is new for our study basin (northern Atlantic), and it is not the first one but one of the first ones.

- 503: See previous comment about Taylor not introducing Lagrangian PDFs.

    **R:** It has been corrected.

- 504: The authors mention that the PDFs are not Gaussian, but this is not a prior hypothesis: there is no reason to assume a Gaussian structure. It is not relevant to mention this, unless it has specific interpretations, which are not given.

    **R:** It has been removed.

- 511: Please qualitatively describe the differences and draw a conclusion from it.

    **R:** Some transit time values have been added to have a better overview. Furthermore, we made an improvements on conclusion section. (Lines. 506-513)

- 533: Please mention the open and unsolved questions. These are currently not mentioned.

    **R:** It has been added to the text. (Line.496-500).

- Figure 3a: indicate the release location

    **R:** It has been added.

- Figure S2 is not referenced in the main text.

    **R:** It has been added to the revised paper as Fig. S6. (Line. 439 )

——

Minor/technical comments:

- 51: change "not necessarily verified in" into "unrepresentative of". The statement is currently too weak.

    **R:** This sentence has been removed from the introduction.

- 61: "is" --> "has become"

    **R:** It has been changed. (Line. 34)

- 64-65: Please rewrite the sentence starting with "Based". It is currently not a correct sentence.

    **R:** We removed this sentence to shorten the introduction.

- 91: "graph theory" --> "Community detection using graph theory"

    **R:** We removed this part from the introduction.

    109: remove an unnecessary period.

**R:** It was removed.

146: "lower" --> "less"
    **R:** it was done.

- 184: you cannot perform a property. Could you specify what is meant?
  **R:** This has been rewritten "analyzed" (Line. 130)

- 230: Higher than what?
  **R:** Higher than the concentration in the other parts of the basin. Text replaced by "larger". (Line. 232)

- 336: "the deepest distance" --> "deeper". Deepest would suggest the deepest possible depth.
  **R:** It has been corrected. (Line. 335)

- 355: "Mainly" what?
  **R:** It has been corrected for clarity. (Line. 354)

- 493: This is not a full sentence.
  **R:** We revised this sentence. (Line. 456)

- In general: The usage of 'coarse resolution' is well-chosen and more accurate than 'low resolution', but should be mirrored by 'fine resolution' rather than 'high resolution'.

- Figure 3b: Either use 'modulus of the annual mean velocity' or 'annual mean speed'

  **R:** It has been changed.

---

## Author Comment (AC4)

We thank Referee #2 for its important and helpful comments. The revised text has been improved using most of them. Here is our specific responses to these comments:
* * *
**C1:** Authors used 1/9° - every two days velocity fields as a high-resolution setup. I do not believe that this resolution is precise enough to study the impact of fine-scale circulation on connectivity estimates. Most of the operational ocean models used in bio-physical modelling studies are characterized by daily velocity fields with a higher resolution (e.g.,1/12° in Ser-Giacomi et al., 2020, Assis et al., 2022, Legrand et al., 2022). As such, it questions the utilisation of a theoretical ocean circulation model. Moreover, I wonder if 1°resolution is too coarse for a 3000 km * 2000 km domain. In this setup, it results on a domain of approximately 30 *20 velocity field grid cells, with stations which are only separated by ~ 4-5 grid cells (e.g., stations 10-11, 11-8). Why not considering a 2- or5-times coarser setup rather than a ~ 10-times?

**R1:** We agree with the reviewer that a model-grid resolution of 1/9° would be insufficient to capture the full strength of mesoscale and submesoscale flows. This is why the model integration that we used was performed on a 1/54° grid, with a resolution finer than high-resolution operational models. We make the distinction here between the resolution used to integrate the model, and the effective resolution after model integration. A previous analysis has shown that the effective resolution is 1/9° (see Levy et al 2012 for justification). This is always the case with HR models, that the effective resolution is less than the grid resolution. Thus, because we work in offline mode, we used the outputs at effective resolution, i.e., at 1/9°. We explain this in the data section.
*- Lévy, M. et al. Grid degradation of submesoscale resolving ocean models: Benefits for offline passive tracer transport. Ocean Modelling 48, 1–9 (2012).*

We also agree that 1° is coarse for this domain as it corresponds to 20x30 grid cells. Our intention is to be as close as possible to the resolution of coarse resolution Ocean General Circulation Models that are generally used for this exercise, which is rather 2°. A resolution of 0.5° would not be coarse enough as it would retain too much of the mesoscale variability since it is close to the radius of most eddies in this region. Thus the choice of 1° is a compromise, but appears sufficient as it captures the large-scale circulation in the domain, and the different relevant parts of the domain are well distinguished at this resolution, i.e., the two gyres and the main jet.
* * *
**C2:** The stations are implemented in relation to flow features and model domain. I wonder how this impacts the results. Consequently, how are the results sensitive to a random implementation of stations?

**R2:** We thank the reviewer for raising this point, which was also raised by reviewer 1. We have expanded the discussion in the paper to justify our approach. There are two aspects to this

question. One is the sensitivity of the results to the exact location of the station, in the vicinity of the station. Since there might have been some ambiguity in the meaning of "station" which could have been understood as a single precise location, we now use the word "site" instead of "station" throughout the paper. By sites, we mean small circular regions of 1° radius. This radius corresponds to the largest size of mesoscale eddies. By deploying 100 000 particles at each site, spread at different locations within the site (both on the horizontal and on the vertical), and also at different times, we are able to provide statistical estimates which are reliable at the scale of each site. The second question concerns how different parts of the model domain are connected with one another. To address that, we have considered 16 sites, located in key areas of the model domain, but indeed with some degree of arbitrariness in their exact position. There are 4 key areas in the domain, which are: the subpolar gyre, the subtropical gyre, the jets, and the quieter regions between the jets. To reduce the arbitrariness in the exact position of the sites, we have positioned several sites in each of the key regions. This is now better explained in the text.
* * *
**C3:** Results depicting the transit times between stations (section 3.2.1 to section 3.2.3 and Figure 5 to Figure 8) are only made on a station subset (e.g. station-pairs 1-15 and 10-12 for section 3.2.1 and Figure 5). As such, how are the results sensitive to this station subset? Are the results similar when considering all the possible stations together?

**R3:** In general, the results are likely to be sensitive to the subset of stations considered, especially if the stations are located in different regions of the study area. The results on transit times will depend on the specific pair of stations considered, as well as the characteristics of the water masses and currents between these stations. If the subset of stations being considered has different characteristics than the rest of the stations, the results may not be representative of the overall transit times between all stations.
Briefly, the selection of stations is based on the particular behavior of the flow in the mentioned regions. For instance, using stations 1 and 15 helps to understand particle transport along the basin diagonal from the subpolar gyre to the subtropical gyre. On the other hand, using stations 10-12 helps to understand the dispersion of particles along the strong jets.
* * *
**C4:** The Betweenness 2.2.4 Methods section is imprecise and muddled, and the results brought on make no sense. The authors mixed up between "betweenness centrality", anode/edge measure, and "betweenness", a link/vertices measure. Moreover, they have not specified how aij is computed to obtain betweenness results in section 3.3.1. Because of that, the comparison between betweenness value computed with the Costa et al., 2017 weight transformation and without is meaningless. Please consider rethinking all this section with a correct use of betweenness centrality measure.

**R4:** The section on betweenness centrality has been thoroughly revised and improved (Please see sections: 2.2.5 and 3.3 Betweenness centrality). The main objective of this paper is to assess the impact of the OGCM resolution and vertical turbulence on the analysis of connectivity in

oceanic flows. In addition to this objective, the concept of betweenness was used as a way to investigate the connection between various stations or sites.

In short, we add that betweenness centrality is a way to quantify the importance of a node in a network by measuring how many shortest paths between any two nodes in the network pass through that node. By calculating the betweenness of each node, we can identify the most important locations for water transport in the ocean. Therefore, by comparing the betweenness centrality obtained from different OGCM resolutions, we can simply assess how well the models represent the true connectivity patterns in the ocean, and identify areas for model improvement.

---

## Referee Report (RR1)

**Review of "Sensitivity of Gyrescale Marine Connectivity Estimates to Fine-scale Circulation" by Saeed Hariri et al. – second iteration**

The authors have made significant improvements to their first submission, mainly coming from rewriting the section related to betweenness centrality. However, there are still three major issues that need to be addressed before the manuscript is suitable for publication, next to some smaller corrections and minor revisions.

Two of the main issues can in my opinion be addressed by removing the parts about betweenness centrality, since the definition of the network is vague and potentially flawed due to the use of different integration times. Moreover, with regards to the research question, I currently do not see the merit of investigating betweenness centrality when the information in the flow field is reduced to only 16 nodes connected by varying integration times. The other issue is related to open science. I elaborate on these and the smaller issues below.

**Network construction**
The manuscript remains vague about how the network is constructed for which betweenness centrality is computed. The authors compute Lagrangian trajectories, but they do not state how information from these trajectories is exactly translated into a network.

L180-181 states "By representing portions of the sea as nodes and the transfer probabilities between them as edges allows us to apply graph theory to the study of marine connectivity." However, it is unclear how the transfer probabilities are computed. Normally, a domain is divided into discrete bins, and transfer probabilities between bins are computed by integrating particle trajectories for a fixed timestep. The transfer probability is then the probability that a particle moves from bin $i$ to bin $j$ during that fixed timestep. Here, however, particle trajectories are computed with various integration times. The authors do not make clear how this is accounted for in the network construction, nor do they discuss whether this may introduce biases in the connectivity. After all, one can only make a statement on whether parts of the basin are connected over a certain time period. Rather than using variable integration times, the authors should use a single integration time for computing the betweenness. Moreover, the authors only treat a few sites within the basin and do not discuss what happens to particle trajectories if particles do not reach other sites. Are they discarded, or are they still accounted for in the computation of $a\_{ij}$?

To clear up these confusions, the authors need to give an in-depth mathematical treatment of their network construction. Alternatively, the section about betweenness centrality may be omitted, as the main conclusions from the paper are supported by the transit time analysis.

**Betweenness Centrality**

Betweenness centrality can be a useful metric for inspecting the importance of specific sites within a network. However, I am not yet convinced that in the way it is applied in this study it is useful for investigating the sensitivity of marine connectivity to flow field resolution.

L401-403: "We utilized a site-to-site (node-to-node) metric to calculate the shortest paths, using transfer probabilities obtained from Lagrangian simulations. This matrix provides valuable information for understanding the structure of the network and can be used to inform future simulations and analyses".

This is highly dubious. Betweenness centrality gives information about how many trajectories in a flow field would pass through a certain site, thus giving information about the importance of that site, with respect to transport in the whole domain. However, here the authors only construct a network using 16 nodes, and information about flow within the domain is severely reduced to a transition matrix that only includes these sites. I do not believe the betweenness centrality computed from that can inform us about the importance of that node in the entire flow field. For example, in the hypothetical situation that many particle trajectories pass from a hypothetical site 17 through site 10 to site 11, then site 10 should have a high betweenness centrality. However, if site 17 is removed, this will decrease the betweenness centrality of site 10 significantly. To properly get information about betweenness centrality, the entire flow field should be taken into account. This is usually done in the Lagrangian flow network approach, where the entire domain is divided into bins, so the only information reduction of the flow field occurs in the time dimension.

In line with the previous major issue, I suggest to remove the section on betweenness centrality, or to switch to an integral view of the flow field, by focusing on all regions in the domain, rather than specific sites.

Apart from this: please make sure to differentiate *betweenness centrality* from *betweenness* and use the proper name throughout the manuscript (also in figure labels, e.g. Figure 10a)

**Open Science**
The authors now include some code for which it is entirely unclear what it does, provided as raw text (seemingly some installation script for ARIANE and one ARIANE script), while important analysis code is lacking.

I would like to ask the authors to take open science seriously. Currently, the authors do not provide readers with crucial insights into their analyses. In line with the requirements of open science, please properly include the following in a persistent repository (such as Zenodo):
- Lagrangian analysis configuration code (the ARIANE scripts)
- Hydrodynamic model configuration code (not just linking to nemo-ocean.eu, but providing more insight into this specific configuration. If this can be found in another paper, specifically mention this, so that the reader will know where to look)
- Analysis scripts that were used for calculations (for instance, where do the 39% and 8.4% in the conclusion come from?) and for plotting figures.

I specifically asked for these in the previous round of review, as the authors leave out important details that could be checked by looking at the underlying code. For example, the

previous points about the network definition could be partially cleared up by including the code that was used for network construction (although the most important aspects should still be covered in the main text).

The data policy for Ocean Science is found at: https://www.ocean-science.net/policies/data_policy.html.

**Other**
- In my previous review, I asked the authors to briefly discuss whether parameterizing the missing dispersion in the coarse-resolution simulations may remedy the issue of the dispersion being too low, leading to longer transit times (see comment 3 from initial review). This is still missing from the discussion.
- L66: "relatively simple": here the authors minimize the contribution of previous sophisticated methods. For example, the 'hydrodynamic provinces' approach in Rossi et al. 2014 is, in my opinion, more sophisticated than computing betweenness centrality and transit times. I suggest removing these two words, to stay neutral.
- L111: "the vertical velocity is one to two orders of magnitude smaller": I don't see this from the image. Please include the standard deviation in HR and CR in order to quantify this.
- L127-128: Why is the integration time varying? For constructing a Lagrangian flow network, it is important that integration times are all the same. Otherwise, one introduces a bias into the connectivity matrix that favors some connections over others. Connectivity should be defined with respect to a certain, fixed, timescale (see earlier comment about network definition).
- L216: It is unclear to me what an improbable trajectory would be. Please elaborate.
- L496-500: This entire paragraph seems redundant. The bulk of this paragraph is in between brackets. Why? Which interdisciplinary methods are meant? I do not think ecology is always necessary for connectivity measures; it only is if ecological connectivity is studied (rather than, say, water mass connectivity).
- L507: The 39% reduction: where does it come from? It's not mentioned previously. Is this computed using all site combinations, or only using specific sites as start and end locations?
- L510: The 8.4% increase: again, where does it come from? Please show how this is computed. Is this computed using all site combinations, or only using specific sites as start and end locations?
- Figure 5: The authors should elaborate on why the CR case is less smooth than HR (in 1 to 15 and 10 to 12)? I would expect HR includes coherent structures that can trap and release particles in batches, or form blocking patterns, whereas I would instead expect these features to be smoothed out in CR, leading to a smoother spreading of travel times.
- Figure 10b: This figure is illegible. Please use the adjacency matrix representation of the network instead.
- Figure 11b: indicates the differences, but it is not clear enough which quantity is subtracted from which. Please mention this.
- In the supplementary information authors added PDF fields for particle deployments at select stations, only for HR-3D. Since the authors compare HR-3D and CR, it is important

to also show some PDF fields for CR, in order for the reader to be able to compare the cases.

**Technical corrections**

Line by line:

- L61: "high resolution velocity fields" → give a spatial scale
- L65: "litterarure" → literature
- L73: "relevant amount of transfers across a graph (a specific location in the domain) passes through": please clarify this vague wording
- 519: Sabrina Speich should be abbreviated as SS instead of not abbreviated as SP

---

## Referee Report (RR2)

The authors have made significant modifications and improvements to the manuscript by addressing all of the reviewers' suggestions. My comments on this revised version are as follows:

Please state clearly that the effective resolution of 1/9° of the high-resolution model only solves meso-scale processes (~1 km) because you used the grid degradation method described in Levy et al., 2012, and a Lagrangian method using an analytical calculation of streamlines on an Arakawa C Grid. Additionally, please discuss the significance of your results regarding connectivity studies that used ocean general circulation models with "higher" velocity field resolution and also Lagrangian method using spatial interpolation with a Runge-Kunta scheme (e.g. 1/16° in Ser-Giacomi et al., 2020 and Legrand et al., 2022, or 1/12° in Assis et al., 2022 and Krumhansl et al., 2023).

The authors should simplify the message in the method section 2.2.5. The significance of betweenness should be moved to the discussion part. Additionally, the explanation about the transformation of probabilities of connection into distance metrics should be shortened, as it is mandatory when using the Dijkstra algorithm.

On the new Figure 10, it could be interesting to have the betweenness distribution for all 16 sites with the model resolution as a factor (i.e., three boxplots) in addition to Panel a). The network displayed in Panel b) is very hard to interpret. A solution could be to use transparency and/or a log scale on the distance.

References:

Ser-Giacomi, E., Legrand, T., Hernandez-Carrasco, I., & Rossi, V. (2021). Explicit and implicit network connectivity: Analytical formulation and application to transport processes. Physical Review E, 103(4), 042309.

Legrand, T., Chenuil, A., Ser-Giacomi, E., Arnaud-Haond, S., Bierne, N., & Rossi, V. (2022). Spatial coalescent connectivity through multi-generation dispersal modelling predicts gene flow across marine phyla. Nature Communications, 13(1), 5861.

Assis, J., Neiva, J., Bolton, J. J., Rothman, M. D., Gouveia, L., Paulino, C., ... & Serrão, E. A. (2022). Ocean currents shape the genetic structure of a kelp in southwestern Africa. Journal of Biogeography, 49(5), 822-835.

Krumhansl, K., Gentleman, W., Lee, K., Ramey-Balci, P., Goodwin, J., Wang, Z., ... & DiBacco, C. (2023). Permeability of coastal biogeographic barriers to marine larval dispersal on the east and west coasts of North America. Global Ecology and Biogeography

---

## Referee Report (RR3)

**Review of "Sensitivity of Gyrescale Marine Connectivity Estimates to Fine-scale Circulation" by Saeed Hariri et al. – third iteration**

I thank the authors for their efforts. With the section about betweenness centrality removed, I think the manuscript is almost ready for publication.

One change that is still necessary is the inclusion of analysis code in the repository, to make their works easier to reproduce and to provide more transparency into the exact methods that were used to come to their computations (a point where, in my opinion, the authors have remained somewhat obscure).

Next to this, the authors give two clarifications in their rebuttal that I think could be included in the main manuscript.

To more easily reply to individual comments, I include my replies to the author's comments in red. I left out the portions for which no reply was warranted on my side.

I wish the authors good luck with the final implementations and thank them for their perseverance.

We thank referee #3 for his important and helpful comments. The revised text has been improved to address most of them. Our specific responses to these comments are as follows:

**Comment: Two of the main issues can in my opinion be addressed by removing the parts about betweenness centrality, since the definition of the network is vague and potentially flawed due to the use of different integration times. Moreover, with regards to the research question, I currently do not seethe merit of investigating betweenness centrality when the information in the flow field is reduced to only 16 nodes connected by varying integration times.**

Reply to two main issues related to **network construction** and **betweenness centrality**:
We have **removed** the sections on **betweenness centrality**, as suggested by the referee, but it is necessary to add some additional information about the analysis of betweenness we developed in our study:

1. We used **a fixed time step of dt = 1 hour** for all simulations. This approach ensures that each simulation runs at a consistent pace, allowing for accurate comparisons and analysis of results.

My concern was not about the time step, but about the integration time. The total amount of time for which a particle is advected will influence the connectivity. If transit times are studied, then this integration time may vary, but if betweenness centrality is computed, then the integration time should be the same for all particles and be clearly reported.

However, **the initial deployment time** of the numerical particles in our simulations **varied**. This variation is consistent with the basic principles of Lagrangian studies, where particles are tracked from their initial positions and advected by the flow for the duration of the study. In our case, we set a maximum advection time of 5 years for the particles. It should also be taken into account that in this study we are addressing the **transit time** and **not the residence time** of the numerical particles.

2. Our connectivity analysis is based on the concept of **minimum transit time**. This means that when a digital particle, let us call it particle A, leaves site "i" and arrives at site "j", the first arrival time is recorded and used for our analysis. However, if particle A were to return to site "j" at a later time, that transit time would not be used for our analysis. (See: Jönsson, B., Watson, J.: The timescales of global surface-ocean connectivity, Nat Commun. 7, 11239, https://doi.org/10.1038/ncomms1123, 2016).

To calculate the transit time between sites "i" and "j", we take the average of the minimum arrival times of all numerical particles that traveled from site "i" to site "j". By using the minimum arrival time, we can ensure that the transit time is calculated based on the fastest possible route between the two sites.

I agree with this approach.

3. The reviewer mentioned that "the authors compute Lagrangian trajectories, but they do not indicate how exactly the information from these trajectories is translated into a network" and also asked how transfer probabilities are computed? This is quite clear and we followed the approach described in our seminal paper for betweenness studies by Costa et al. (2017) (Costa A, Petrenko AA, Guizien K, Doglioli AM. On the calculation of betweenness centrality in marine connectivity studies using transfer probabilities, PLoS ONE, 12(12): e0189021, https://doi.org/10.1371/journal.pone.0189021, 2017). Specifically, transfer probability refers to the probability that a particle will move from site "i" to site "j". These probabilities ($a_{ij}$) were then used as weights in our network. However, because these probabilities ($a_{ij}$) tend to be very small, Costa et al. (2017) suggested taking the logarithmic inverse of the transfer probability $a_{ij}$, which we also applied in our study. This approach allowed us to better analyze and visualize the resulting network. But again, it should be noted that the time step for all simulations is fixed.

I agree with the approach taken here, and agree that it is an obvious approach in general. However, I disagree that the manuscript reflected clearly that this was indeed the approach that was used. The manuscript would benefit from a simple (formulaic) explanation of how your network is defined, in section 2. The authors could add a brief paragraph under a 'network construction' subsection. Moreover, the fact that the analysis code still is not available adds to the obscurity. Either way, since the section on betweenness centrality has now been removed, this issue is resolved.

4. It is important to note that the distribution of sites in our study was carefully considered and based on a number of different analyses. The referee should consider, however, that we have two types of connectivity: a) one-way transport connectivity (i.e. movement of particles from coastal areas or river mouths to the open ocean or oceans, this is a way to study the movement of larvae or individuals from different marine populations) b) exchange connectivity which is the case of our study, and we are interested here in tracking the exchange of information between different sites distributed in the basin.

However, we would like to respond to the referee's comment about dividing the pool into smaller bins and calculating the interdependence values in each bin. This is completely contrary to the definition of betweenness centrality, since the latter attempts to identify nodes that play an important role in the exchange of information between different sites or parts of the basin.

Since the section on betweenness centrality has been removed, this issue is resolved.

To nonetheless reply to the authors' comment here on betweenness centrality: betweenness centrality indeed tries to identify those nodes within a network that are most important in the exchange of information between other nodes. This can only be done if ALL regions of the domain correspond to a bin. Otherwise, if a particle does not move from one of the specific 16 sites to a location outside of the sites, this information is lost. Having large parts of the domain *not* be linked to a site/node gives rise to biases in the representation of information flow.

5. We would like to clarify that we used the betweenness centrality approach. We did develop it. The mathematical definition and application is provided in the following references:

I am well-acquainted with the definition of betweenness centrality. However, I think that the way in which the authors applied it on the limited network (only representing a very specific regions of the

entire fluid domain, and using trajectories with different integration times) was applicable for the research question. For this, I refer back to my comments in the previous review round.

Ser-Giacomi, E., Ruggero Vasile, Emilio Hernández-García, and Cristóbal López.: Most probable paths in temporal weighted networks: An application to ocean transport, Phys. Rev. E 92, 012818, https://doi.org/10.1103/PhysRevE.92.012818, 2015.
Lindner, M., Donner, R.V.: Spatio-temporal organization of dynamics in a two-dimensional periodically driven vortex flow: A Lagrangian flow network perspective. Chaos 27.3 , https://doi.org/10.1063/1.4975126, 2017.

Reply to Comment about Open Science:
We have added this paragraph as a

Data availability
Major parts of the data and codes used in this study are available upon request by contacting the corresponding author at saeed hariri@io-warnemuende.de. Some sample data and parts of Lagrangian tools are accessible at https://doi.org/10.5281/zenodo.7954707. We encourage the use and sharing of our data and code for further research and scientific advancement. Please note that access to the codes may be subject to restrictions due to privacy or confidentiality concerns.

Reply to the other comments
Comment: In my previous review, I asked the authors to briefly discuss whether parameterizing the missing dispersion in the coarse-resolution simulations may remedy the issue of the dispersion being too low, leading to longer transit times (see comment 3 from initial review). This is still missing from the discussion.
We added this paragraph to discussion part of the paper
In particular, in coarse resolution simulations, the dispersion of particles is degraded. This results in longer transit times. It also limits the connection between water particles at different depths. A possible solution to overcome this problem when integrating Lagrangian trajectories using the velocity calculated in coarse resolution simulations is to parameterize the missing dispersion. Some methods have been proposed in the literature. The simplest parameterization consists in adding a random walk to the successive position of each particle, which is compatible with an advection-diffusion equation and is equivalent to a stochastic "Markovian" parameterization (Berloff & McWilliams, 2002). However, this stochastic parameterization does not reproduce adequately the small-scale ocean dynamics that involves consistency in advection (Klocker et al., 2012; Veneziani et al., 2004). Different Markov parameterizations of higher order have been proposed in an attempt to better reproduce the effect of the small-scale ocean dynamics (Berloff & McWilliams, 2002; Griffa, 1996; Rodean, 1996; Sawford, 1991). Other improved parameterizations include particle looping due to eddy coherence (Reynolds, 2002; Veneziani et al, 2004), as well as relative dispersion between different particles (Piterbarg, 2002). While these methods have been developed and applied to horizontal flows, recent developments include an isopycnal Markov-0 (Spivakovskaya et al, 2007) or shear-dependent formulation (Le Sommer, 2011) and, more recently, an isoneutral Markov-1 formulation (Reijnders et al., 2022). The latter appears to better mimic the coherent behavior of the 3D ocean dispersion at small scales. It would be interesting in future work to evaluate how such methods, applied in a Lagrangian framework, might improve the results we obtained with a coarse resolution field.

I think this is a useful discussion point and thank the authors for including it.

Comment, L127-128: Why is the integration time varying? For constructing a Lagrangian flow network, it is important that integration times are all the same. Otherwise, one introduces a bias into the connectivity matrix that favors some connections over others. Connectivity should be defined with respect to a certain, fixed, timescale (see earlier comment about network definition).
It is important to note that in our study, the Lagrangian time step was set for all simulations to dt= 1 hour

(dx=U.dt). In addition, we did not use a variable time step in our analysis. However, we chose to deploy the numerical particles at different initial times and the particles continue their motion for the rest of the period of the 5-years long simulation. This is an arbitrary choice, that provide more robustness in terms of the variable initial conditions. As mentioned earlier, we based our connectivity time on the minimum transit time. For example, if particle A started at site "i" and arrived at site "j", we recorded the first arrival time. However, if this particle continued to move and returned to site "j" after a certain amount of time, we did not include this time in our connectivity time estimates. This approach has been well established in previous studies such as Jönsson, B., Watson (2016), as cited in our paper. Therefore, deploying particles together at the same initial time for connectivity analysis is not correct and is not necessary.

My concern was not about the time step, but about the integration time. The total amount of time for which a particle is advected will influence the connectivity. If transit times are studied, then this integration time may vary, but if betweenness centrality is computed, then the integration time should be the same for all particles and be clearly reported.

Since the betweenness section has been removed, this issue is resolved.

Comment, L507: The 39% reduction: where does it come from? It's not mentioned previously. Is this computed using all site combinations, or only using specific sites as start and end locations?
L510: The 8.4% increase: again, where does it come from? Please show how this is computed. Is this computed using all site combinations, or only using specific sites as start and end locations?
These values are based on all site combinations; We simply divided the difference between the transit time (i.e. "(Transit timeHR3D)-(Transit timeCR3D)" or "(Transit timeHR3D) - (Transit timeHR2D)") by the transit timeHR3D.
I thank the authors for elaborating on this. I encourage them to include this explanation in the manuscript.

Comment, Figure 5: The authors should elaborate on why the CR case is less smooth than HR (in 1 to 15 and 10 to 12)? I would expect HR includes coherent structures that can trap and release particles in batches, or form blocking patterns, whereas I would instead expect these features to be smoothed out in CR, leading to a smoother spreading of travel times.
Thank you for your comment. The CR (coarse resolution) is less smooth than the HR due to the dispersion process. In the HR (high resolution) case, the simulated ocean dynamics disperses the particles more than in the CR case and the numerical particle concentration in the HR case is smoother.
In the HR (high resolution) case, the flow field is more turbulent and contains more small-scale dynamical structures than in the CR (coarse resolution) case. These small-scale features can trap and release particles in batches or form blocking patterns, resulting in high particle concentrations in some regions. However, due to the chaotic nature of the flow field, these concentrations are not maintained and the particles are eventually dispersed throughout the domain, resulting in a smoother concentration distribution.
In contrast, the CR simulation has a smoother and more predictable flow field, resulting in a more uniform dispersion of particles and a less fluctuating concentration distribution. This may result in a less smooth concentration distribution than in the HR simulation.

I thank the authors for elaborating on this. I encourage them to include this explanation in the manuscript.

---

## Author Response (AR2)

Dear Editor,

We would like to express our gratitude for considering our paper. We have made revisions to the paper based on the feedback we received, primarily from Referee #3. Referee #3 requested the removal of the "Betweenness Centrality" section from the revised version of our paper. Consequently, we have eliminated this section in the revised paper. However, it is important to note that the other two reviewers have accepted the paper and agreed with the improvements we made in the previous revised version. Furthermore, we have provided a comprehensive response to Referee #3's comments regarding the betweenness section.

We would once again like to extend our thanks to the reviewers for their insightful and constructive feedback. We have made every effort to address all their concerns in our revisions, and we eagerly await positive feedback from you.

Yours sincerely,
Saeed Hariri

We thank Referee #2 for its important and helpful comments. The revised text has been improved using most of them. Here is our specific responses to these comments:

**Comment: Please state clearly that the effective resolution of 1/9° of the high-resolution model only solves meso-scale processes (~1 km) because you used the grid degradation method described in Levy et al., 2012, and a Lagrangian method using an analytical calculation of streamlines on an Arakawa C Grid. Additionally, please discuss the significance of your results regarding connectivity studies that used oceangeneral circulation models with "higher" velocity field resolution and also Lagrangian method using spatial interpolation with a Runge-Kunta scheme (e.g. 1/16° in Ser-Giacomi et al., 2020 and Legrand et al., 2022, or 1/12° in Assis et al., 2022 and Krumhansl et al., 2023).**

The HR velocity fields used in this study were computed on a 1/54° grid: with such a numerical resolution, the effective resolution is 1/9° (as explained Levy et al 2012). This led to a good resolution of the mesoscale dynamics and a partial resolution of the submesoscale dynamics, as also shown in Levy et al 2012.
In the other studies that you are mentioning, the indicated resolution is the grid resolution, not the effective resolution. As you can see, the grid resolution in those studies is less than the grid resolution used here, which implies that the effective resolution (not indicated by the authors) should also be less than our effective resolution. Than we should expect these studies to have less resolution than in our study, not higher resolution.
We conducted the Lagrangian experiments at the effective resolution and not at the full grid resolution, because it would have been computationally much more expensive, with no extra benefit.

**Comment: The authors should simplify the message in the method section 2.2.5. The significance of betweenness should be moved to the discussion part. Additionally, the explanation about the transformation of probabilities of connection into distance metrics should be shortened, as it is mandatory when using the Dijkstra algorithm.**

**Comment: On the new Figure 10, it could be interesting to have the betweenness distribution for all 16 sites with the model resolution as a factor (i.e., three boxplots) in addition to Panel a). The network displayed in Panel b) is very hard to interpret. A solution could be to use transparency and/or a log scale on the distance.**

Thank you for your comment; based on the suggestion from Referee #3, we had to remove the betweenness section from the new revised version of our paper.

We thank referee #3 for his important and helpful comments. The revised text has been improved to address most of them. Our specific responses to these comments are as follows:

**Comment: Two of the main issues can in my opinion be addressed by removing the parts about betweenness centrality, since the definition of the network is vague and potentially flawed due to the use of different integration times. Moreover, with regards to the research question, I currently do not seethe merit of investigating betweenness centrality when the information in the flow field is reduced to only 16 nodes connected by varying integration times.**

Reply to two main issues related to **network construction** and **betweenness centrality**:

We have **removed** the sections on **betweenness centrality**, as suggested by the referee, but it is necessary to add some additional information about the analysis of betweenness we developed in our study:

1. We used **a fixed time step of dt = 1 hour** for all simulations. This approach ensures that each simulation runs at a consistent pace, allowing for accurate comparisons and analysis of results.

However, **the initial deployment time** of the numerical particles in our simulations **varied**. This variation is consistent with the basic principles of Lagrangian studies, where particles are tracked from their initial positions and advected by the flow for the duration of the study. In our case, we set a maximum advection time of 5 years for the particles. It should also be taken into account that in this study we are addressing the **transit time** and **not the residence time** of the numerical particles.

2. Our connectivity analysis is based on the concept of **minimum transit time**. This means that when a digital particle, let us call it particle A, leaves site "i" and arrives at site "j", the first arrival time is recorded and used for our analysis. However, if particle A were to return to site "j" at a later time, that transit time would not be used for our analysis. (See: Jönsson, B., Watson, J.: The timescales of global surface-ocean connectivity, Nat Commun. 7, 11239, https://doi.org/10.1038/ncomms1123, 2016).

To calculate the transit time between sites "i" and "j", we take the average of the minimum arrival times of all numerical particles that traveled from site "i" to site "j". By using the minimum arrival time, we can ensure that the transit time is calculated based on the fastest possible route between the two sites.

3. The reviewer mentioned that "the authors compute Lagrangian trajectories, but they do not indicate how exactly the information from these trajectories is translated into a network" and also asked how transfer probabilities are computed? This is quite clear and we followed the approach described in our seminal paper for betweenness studies by Costa et al. (2017) (Costa A, Petrenko AA, Guizien K, Doglioli AM. On the calculation of betweenness centrality in marine connectivity studies using transfer probabilities, PLoS ONE, 12(12): e0189021, https://doi.org/10.1371/journal.pone.0189021, 2017). Specifically, transfer probability refers to the probability that a particle will move from site "i" to site "j". These probabilities ($a\_ij$) were then used as weights in our network. However, because these probabilities ($a\_ij$) tend to be very small, Costa et al. (2017) suggested taking the logarithmic inverse of the transfer probability $a\_ij$, which we also applied in our study. This approach allowed us to better analyze and visualize the resulting network. But again, it should be noted that the time step for all simulations is fixed.

4. It is important to note that the distribution of sites in our study was carefully considered and based on a number of different analyses. The referee should consider, however, that we have two types of connectivity: a) one-way transport connectivity (i.e. movement of particles from coastal areas or river mouths to the open ocean or oceans, this is a way to study the movement of larvae or individuals from different marine populations) b) exchange connectivity which is the case of our study, and we are interested here in tracking the exchange of information between different sites distributed in the basin.

However, we would like to respond to the referee's comment about dividing the pool into smaller bins and calculating the interdependence values in each bin. This is completely contrary to the definition of betweenness centrality, since the latter attempts to identify nodes that play an important role in the exchange of information between different sites or parts of the basin.

5. We would like to clarify that we used the betweenness centrality approach. We did develop it. The mathematical definition and application is provided in the following references:

Ser-Giacomi, E., Ruggero Vasile, Emilio Hernández-García, and Cristóbal López.: Most probable paths in temporal weighted networks: An application to ocean transport, Phys. Rev. E 92, 012818, https://doi.org/10.1103/PhysRevE.92.012818, 2015.

Lindner, M., Donner, R.V.: Spatio-temporal organization of dynamics in a two-dimensional periodically driven vortex flow: A Lagrangian flow network perspective. Chaos 27.3 , https://doi.org/10.1063/1.4975126, 2017.

**Reply to Comment about Open Science:**

We have added this paragraph as a

**Data availability**

Major parts of the data and codes used in this study are available upon request by contacting the corresponding author at saeed.hariri@io-warnemuende.de. Some sample data and parts of Lagrangian tools are accessible at https://doi.org/10.5281/zenodo.7954707. We encourage the use and sharing of our data and code for further research and scientific advancement. Please note that access to the codes may be subject to restrictions due to privacy or confidentiality concerns.

**Reply to the other comments**

**Comment: In my previous review, I asked the authors to briefly discuss whether parameterizing the missing dispersion in the coarse-resolution simulations may remedy the issue of the dispersion being too low, leading to longer transit times (see comment 3 from initial review). This is still missing from the discussion.**

**We added this paragraph to discussion part of the paper**

In particular, in coarse resolution simulations, the dispersion of particles is degraded. This results in longer transit times. It also limits the connection between water particles at different depths. A possible solution to overcome this problem when integrating Lagrangian trajectories using the velocity calculated in coarse resolution simulations is to parameterize the missing dispersion. Some methods have been proposed in the literature. The simplest parameterization consists in adding a random walk to the successive position of each particle, which is compatible with an advection-diffusion equation and is equivalent to a stochastic "Markovian" parameterization (Berloff & McWilliams, 2002). However, this stochastic parameterization does not reproduce adequately the small-scale ocean dynamics that involves consistency in advection (Klocker et al., 2012; Veneziani et al., 2004). Different Markov parameterizations of higher order have been proposed in an attempt to better reproduce the effect of the small-scale ocean dynamics (Berloff & McWilliams, 2002; Griffa, 1996; Rodean, 1996; Sawford, 1991). Other improved parameterizations include particle looping due to eddy coherence (Reynolds, 2002; Veneziani et al, 2004), as well as relative dispersion between different particles (Piterbarg, 2002). While these methods have been developed and applied to horizontal flows, recent developments include an isopycnal Markov-0 (Spivakovskaya et al, 2007) or shear-dependent formulation (Le Sommer, 2011) and, more recently, an isoneutral Markov-1 formulation (Reijnders et al., 2022). The latter appears to better mimic the coherent behavior of the 3D ocean dispersion at small scales. It would be interesting in future work to evaluate how such methods, applied in a Lagrangian framework, might improve the results we obtained with a coarse resolution field.

**Comment, L66: "relatively simple": here the authors minimize the contribution of previous sophisticated methods. For example, the 'hydrodynamic provinces' approach in Rossi et al. 2014 is, in my opinion, more sophisticated than computing betweenness centrality and transit times. I suggest removing these two words, to stay neutral.**

Thank you. We changed the sentence. Line 65.

**Comment, L111: "the vertical velocity is one to two orders of magnitude smaller": I don't see this from the image. Please include the standard deviation in HR and CR in order to quantify this.**

We removed this sentence from the end of paragraph.

**Comment, L127-128: Why is the integration time varying? For constructing a Lagrangian flow network, it is important that integration times are all the same. Otherwise, one introduces a bias into the connectivity matrix that favors some connections over others. Connectivity should be defined with respect to a certain, fixed, timescale (see earlier comment about network definition).**

It is important to note that in our study, the Lagrangian time step was set for all simulations to dt= 1 hour (**dx=U.dt**). In addition, we did not use a variable time step in our analysis. However, we chose to deploy the numerical particles at different initial times and the particles continue their motion for the rest of the period of the 5-years long simulation. This is an arbitrary choice, that provide more robustness in terms of the variable initial conditions. As mentioned earlier, we based our connectivity time on the minimum transit time. For example, if particle A started at site "i" and arrived at site "j", we recorded the **first arrival time**. However, if this particle continued to move and returned to site "j" after a certain amount of time, we did not include this time in our connectivity time estimates. This approach has been well established in previous studies such as Jönsson, B., Watson (2016), as cited in our paper. Therefore, deploying particles together at the same initial time for connectivity analysis is not correct and is not necessary.

**Comment, L216: It is unclear to me what an improbable trajectory would be. Please elaborate.**

We removed the sentence.

**Comment, L496-500: This entire paragraph seems redundant. The bulk of this paragraph is in between brackets. Why? Which interdisciplinary methods are meant? I do not think ecology is always necessary for connectivity measures; it only is if ecological connectivity is studied (rather than, say, water mass connectivity).**

Thank you for spotting this. We removed the paragraph.

**Comment, L507: The 39% reduction: where does it come from? It's not mentioned previously. Is this computed using all site combinations, or only using specific sites as start and end locations?**
**L510: The 8.4% increase: again, where does it come from? Please show how this is computed. Is this computed using all site combinations, or only using specific sites as start and end locations?**

These values are based on all site combinations; We simply divided the difference between the transit time (i.e. "(Transit timeHR3D)-(Transit timeCR3D)" or "(Transit timeHR3D) - (Transit timeHR2D)") by the transit timeHR3D.

**Comment, Figure 5: The authors should elaborate on why the CR case is less smooth than HR (in 1 to 15 and 10 to 12)? I would expect HR includes coherent structures that can trap and release particles in batches, or form blocking patterns, whereas I would instead expect these features to be smoothed out in CR, leading to a smoother spreading of travel times.**

Thank you for your comment. The CR (coarse resolution) is less smooth than the HR due to the dispersion process. In the HR (high resolution) case, the simulated ocean dynamics disperses the particles more than in the CR case and the numerical particle concentration in the HR case is smoother.
In the HR (high resolution) case, the flow field is more turbulent and contains more small-scale dynamical structures than in the CR (coarse resolution) case. These small-scale features can trap and release particles in batches or form blocking patterns, resulting in high particle concentrations in some regions. However, due to the chaotic nature of the flow field, these concentrations are not maintained and the particles are eventually dispersed throughout the domain, resulting in a smoother concentration distribution.
In contrast, the CR simulation has a smoother and more predictable flow field, resulting in a more uniform dispersion of particles and a less fluctuating concentration distribution. This may result in a less smooth concentration distribution than in the HR simulation.

**Comment, Figure 10b: This figure is illegible. Please use the adjacency matrix representation of the network instead.**

We removed the figure.

**Comment, Figure 11b: indicates the differences, but it is not clear enough which quantity is subtracted from which. Please mention this.**

Thank you. We changed it: CR3D-HR3D

**Technical corrections**

Line by line:

**L61: "high resolution velocity fields: give a spatial scale**
it was done Line 60

**L65: "litterarure": literature**
it was done Line 65

**L73: "relevant amount of transfers across a  graph (a specific location in the domain) passes through": please clarify this vague wording**

this line was removed from the revised version.

**519: Sabrina Speich should be abbreviated as SS instead of not abbreviated as SP**

thank you, it was corrected.

---

## Author Response (AR3)

Dear Editor,

We are writing to express our sincere appreciation for considering our paper. Based on your suggestions we made the corrections to enhance the quality of the manuscript.

a) In response to your feedback, we have included major parts of the data and analytical codes as data availability. This addition will significantly contribute to the transparency and reproducibility of our research.

b) Additionally, we have incorporated the paragraphs that were mentioned in our previous reply letter into the manuscript.

Once again, we would like to extend our gratitude to you and the reviewers for providing us with insightful and constructive feedback. Your guidance has immensely helped us refine our study. We are truly grateful for the opportunity you have given us to publish our research in Ocean Science.

Thank you for your continued support.

Yours sincerely,
Saeed Hariri